# Spike-based Neuromorphic Model for Sound Source Localization

**Dehao Zhang**[1,‡]**, Shuai Wang**[1,‡]**, Ammar Belatreche**[2]**, Wenjie Wei**[1]**, Yichen Xiao**[1]**,**
**Haorui Zheng**[3]**, Zijian Zhou**[1]**, Malu Zhang**[1]*** **Yang Yang**[1]

[1] University of Electronic Science and Technology of China
2 Northumbria University
3 Peking University
{zhangdh,wangshuai718}@std.uestc.edu.cn, maluzhang@uestc.edu.cn

## Abstract

Biological systems possess remarkable sound source localization (SSL) capabilities
that are critical for survival in complex environments. This ability arises from the
collaboration between the auditory periphery, which encodes sound as precisely
timed spikes, and the auditory cortex, which performs spike-based computations.
Inspired by these biological mechanisms, we propose a novel neuromorphic SSL
framework that integrates spike-based neural encoding and computation. The
framework employs Resonate-and-Fire (RF) neurons with a phase-locking coding
(RF-PLC) method to achieve energy-efficient audio processing. The RF-PLC
method leverages the resonance properties of RF neurons to efficiently convert
audio signals to time-frequency representation and encode interaural time difference
(ITD) cues into discriminative spike patterns. In addition, biological adaptations
like frequency band selectivity and short-term memory effectively filter out many
environmental noises, enhancing SSL capabilities in real-world settings. Inspired
by these adaptations, we propose a spike-driven multi-auditory attention (MAA)
module that significantly improves both the accuracy and robustness of the proposed
SSL framework. Extensive experimentation demonstrates that our SSL framework
achieves state-of-the-art accuracy in SSL tasks. Furthermore, it shows exceptional
noise robustness and maintains high accuracy even at very low signal-to-noise
ratios. By mimicking biological hearing, this neuromorphic approach contributes
to the development of high-performance and explainable artificial intelligence
systems capable of superior performance in real-world environments.

## 1 Introduction

Sound source localization (SSL) [11, 43] is a critical skill for mammals that enables them to identify
and locate external auditory stimuli. This skill plays a vital role in survival behaviors like prey
detection and predator evasion. Over decades of scientific exploration [39, 48], SSL has evolved
from a purely biological concept to a sophisticated technology with a wide range of applications
across various fields [10, 39]. Today, SSL methods are finding increasing use in areas like security
monitoring [12], robotic navigation [37], and autonomous driving [13, 36].

Early SSL approaches rely on hand-crafted analysis of speech signals from multiple receivers. While
offering a basic ability to localize sound sources, these methods suffer from limitations in accuracy
and robustness. The emergence of Deep Neural Networks (DNNs) and their success in various
domains lead researchers to explore their application in SSL tasks, achieving significant performance
improvements [64, 67]. However, DNN-based approaches face two key challenges. Firstly, DNNs

---

*‡ Equal contribution, * Corresponding author

achieving high SSL accuracy often require substantial computational resources, leading to increased energy consumption. Secondly, DNNs struggle to learn the intricate relationships between localization behaviors and noisy environmental constraints. These limitations hinder the development of portable, edge-based SSL models [16] for real-world environments.

Recently, Spiking Neural Networks (SNNs) [14, 20, 34], inspired by brain neural architectures, have gained significant attention for their energy-efficient simulation of neural systems. Spiking neurons [33] simulate the information transmission mechanism of biological neurons, computing only upon the arrival of input spikes and remaining silent otherwise [53]. Such an event-driven mechanism results in sparser information transmission [60, 61], hence reducing computational costs [6, 69]. Therefore, the SNN-based SSL models enable a more energy-efficient emulation of biological SSL processes. Pan et al. [40] propose a SNN-based SSL model that achieves localization in real audio signals. Chen et al.[7] improve the model's performance through a hybrid encoding method, achieving competitive results with less energy consumption. Although these examples achieve edge-friendly SSL ability, limitations still exist in neural encoding efficiency and robustness under noisy conditions.

In terms of neural encoding, most methods still rely on Fourier Transform (FT) [27, 55] to encode ITD [11] present in the received audio signals into spike trains for processing by back-end SSL model. However, FT operations involve many multiply-accumulate (MAC) computations and require significant computational resources [58] which hinders our goal of developing energy-efficient SSL models. In terms of robustness, the superior localization ability in biology not only relies on ITD cues but also on various auditory mechanisms [22, 54], such as frequency preferences, short-term memory, etc. Frequency preference significantly mitigates the impact of complex environments on localization accuracy [31, 62], and auditory short-term memory effectively filters out irrelevant noise [49, 50], focusing on important auditory signals. However, most SNN-based solutions [3, 32] primarily focus on ITD cues, with little attention to multiple auditory mechanisms. Therefore, investigating more energy-efficient and robust SNN-based SSL models remains a pressing challenge to address.

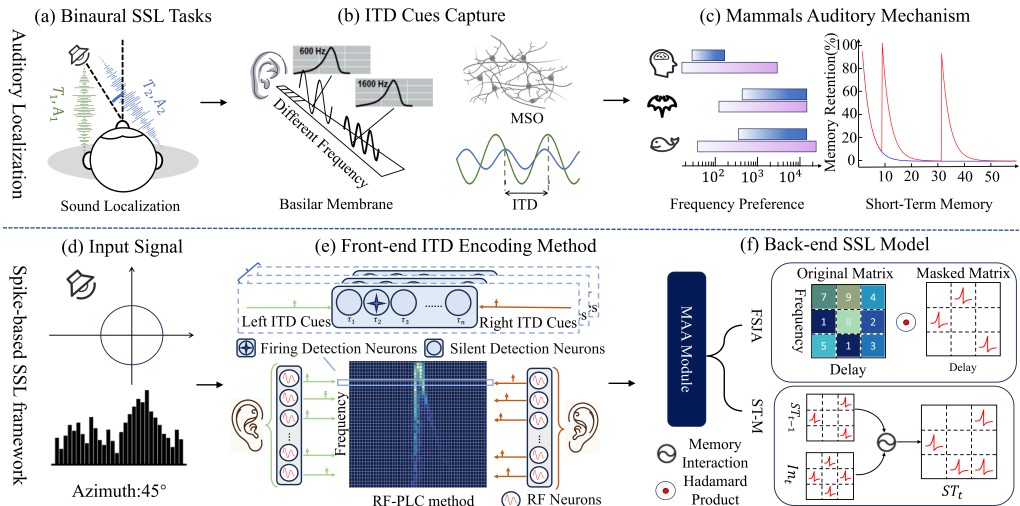

Figure 1: A spike-based SSL framework inspired by biological auditory localization. (a) Schematic of binaural SSL tasks. (d) Simulation of the SSL tasks. The upper section illustrates two key processes involved in mammalian localization: (b) the Basilar Membrane and Medial Superior Olive (MSO) collaborate to capture ITD cues; (c) multiple auditory attention mechanisms further process these ITD cues for precise localization. The lower section presents our spike-based SSL framework, comprising two main components: (e) a front-end ITD encoding method employing RF and detection neurons; (f) A back-end SSL model utilizing multi-auditory attention.

In this paper, we propose a novel SNN-based SSL framework, which primarily comprises an ITD encoding front-end method and a biomimetic localization back-end model. As illustrated in Fig.1, we introduce a phase-locking coding (RF-PLC) method using Resonate-and-Fire (RF) neurons [8, 38] and detection neurons [40]. It simulates the frequency band decomposition function of the basilar membrane and captures ITD cues, respectively. Furthermore, we introduce a novel back-end SSL model based on multi-auditory attention (MAA) that integrates frequency preferences and short-term

memory characteristics. This approach significantly improves both the accuracy and robustness of localization. Extensive experiment on the HRTF [57], Single Words [30], and SLoClas [42] datasets demonstrates that our SNN-based model achieves state-of-the-art performance. Moreover, evaluation in noisy environments reveals the model's enhanced adaptability to real-world conditions. The work introduces the following key contributions:

- **Spike-based ITD encoding**: The RF-PLC method leverages the resonance properties of RF neurons to perform energy-efficient auditory time-frequency transformations, avoiding the high resource costs of FT operations. Additionally, it utilizes a phase-locking loop and ITD detection neurons to encode auditory ITD cues into spike patterns directly, ensuring the fully spike-driven nature of the entire SSL framework.
- **Biologically inspired attention**: The MAA module incorporates knowledge of biological auditory frequency preferences and short-term memory characteristics. Frequency preferences effectively mask the ITD information of irrelevant frequency bands and spatial regions, while short-term memory focuses on the interaction of information across adjacent time steps. This combination enhances the robustness of the SNN model in noisy environments.
- **State-of-the-art performance with reduced complexity**: By integrating these methods, we present a SNN-based SSL framework that achieves state-of-the-art performance while reducing energy consumption to existing works. Additionally, extensive experimentation demonstrates that our system exhibits superior robustness in noisy environments.

## 2 Related Work

### 2.1 ITD Cues for SSL Tasks

To develop biologically inspired models for SSL tasks, researchers have drawn upon the auditory localization mechanisms observed in mammals [23, 25]. The cues of ITD are recognized as critical for these models [29, 44, 46, 47]. ITD refers to the temporal disparity in sound arrival between the ears. Specifically, when a sound source is closer to the listener's right side, audio reaches the right ear sooner than the left. The Jeffress model [3] and BiSoLaNN [56] encode ITD cues into spike trains and corroborate their biological credibility through experiments on barn owls [5]. But these approaches primarily focus on the localization of pure tones, which significantly differs from the time-varying audio signal in daily life. Substantially, some researchers [7, 40] have utilized complex FT operations to obtain the phase information of audio signals. However, FT operations require substantial computational resources and pose significant challenges when implementing systems on edge devices with limited computational capabilities. Therefore, the exploration of low-power ITD encoding methods become a pressing direction to pursue.

### 2.2 Biological Adaptation in Auditory System

In the field of auditory science, frequency preference and short-term memory characteristics are essential for understanding auditory processing. Numerous studies [21, 52, 63] have demonstrated that biological auditory systems exhibit heightened sensitivity to specific frequency ranges, such as 20-20 000 Hz in humans, with other species like bats and blue whales adapted to different ranges. Further research [51] has revealed tonotopic maps and variations in frequency tuning across regions, underscoring the importance of frequency selectivity in hearing. Electrophysiological experiments [31] confirmed that inner hair cells on the basilar membrane of the cochlea exhibit significant differences in response to various frequency bands.

These studies underscore the irreplaceable role of frequency band preference in auditory decision-making. Additionally, compared to visual short-term memory, auditory short-term memory [2, 28] has a shorter retention span. Nonetheless, it is essential for real-time integration and coherent environmental perception. Simultaneously, some researchers [50] propose that neurofeedback training targeting auditory short-term memory can significantly enhance selective attention to auditory signals in noisy environments. Moreover, Zhong et al. [70] suggested that auditory short-term memory can highlight sound source characteristics under reverberant conditions, reducing interference from other sources. However, current SSL methods mainly focus on ITD cues, neglecting these well-established biological mechanisms. Therefore, the effective integration of diverse auditory attention mechanisms within SSL tasks to enhance robustness remains a significant ongoing challenge.

## 3  Method

In this section, we introduce our spike-based SSL framework, which includes a front-end ITD encoding method and a back-end localization model. For the front-end ITD encoding, we propose the energy-efficient RF-PLC method, which uses RF neurons to capture ITD cues and detection neurons to convert these encoded cues into spike patterns. For the back-end localization model, we take inspiration from biological adaptation and propose the MAA mechanism to enhance the model's localization performance and robustness.

### 3.1  RF Phase-locking Coding: A Direct Font-end ITD Encoding Method

Due to the physical separation of the ears, sound waves arrive at each ear with slightly different timing. It leads to differences in the initial phase information between the two audio channels. Pan et al. [40] propose a Multi-Tones Phase Coding (MTPC) method that utilizes this information to exploit ITD cues. However, this approach relies on computationally expensive FT operations and introduces an additional phase transformation step during processing. To overcome these limitations, we propose the RF-PLC method, leveraging RF neurons' resonance filtering and periodic decay properties. This approach effectively eliminates the need for energy-intensive FT and phase transformation processes. Subsequently, a set of detection neurons with varying delays is employed to efficiently encode the ITD cues from different microphones into spike patterns.

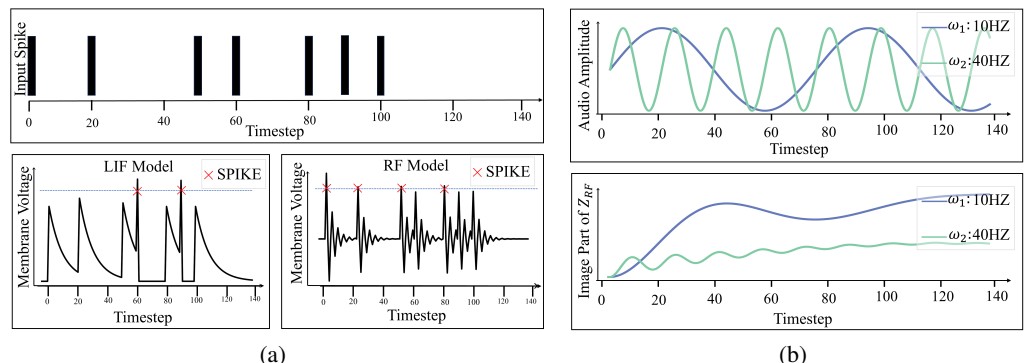

(a)                                                                 (b)

Figure 2: Properties of spiking neuron models. (a) Responses of the LIF and RF neurons to an identical input spike train. We can observe distinct patterns in both membrane voltage accumulation and spiking behavior between the two neuron models. (b) The frequency-selective properties of RF neurons. RF neurons with a resonant frequency of 10 Hz ($\omega = 10$) have a significantly stronger response at 10 Hz compared to the response at 40 Hz.

The first step of our model processes the raw audio to capture ITD cues. We segment the audio into $y_l$ based on the smallest durations by the human ear. These segments are then encoded by specialized RF neurons [38] tuned to different frequency bands. The dynamics of these RF neurons can be described as follows:

$$\mathcal{Z}_{\mathrm{RF}}\left[t\right] = \lambda e^{\mathrm{i}\omega\Delta t}\mathcal{Z}_{\mathrm{RF}}\left[t-1\right] + I\left[t\right], \tag{1}$$

where $\omega$ represents the resonant frequency of the neuron, indicating the number of radians it progresses per second. $\lambda$ serves as the dampening factor, which causes the oscillation to decay exponentially. $\Delta t$ represents the sampling rate, which is set to 1. $I[t]$ denotes the audio input. $\mathcal{Z}_{\mathrm{RF}}[t]$ can be reformulated as $x + \mathrm{i}y \in \mathbb{C}$. A detailed process can be found in Appendix. A. The real component $x$ of $\mathcal{Z}_{\mathrm{RF}}$ reflects the current-like behavior of the neuron, capturing the dynamics of voltage-gated and synaptic currents. The imaginary component $y$ of $\mathcal{Z}_{\mathrm{RF}}[t]$ serves as a voltage-like variable.

Based on Eq. 1, we depict the spiking behavior of the RF neuron in Fig. 2(a) and summarize its characteristics in two aspects. Firstly, the complex form of the RF neuron's dynamics enables it to capture the phase information in a specific frequency band $\omega$, termed resonance filtering. Secondly, the dampening factor $\lambda$ allows it to exhibit periodic decay characteristics when there is no input.

By leveraging RF neurons' resonance filtering and periodic decay properties, we encode input signals into ITD cues efficiently and effectively. To better describe the RF-PLC process, we decompose the

dynamics of the RF neurons into two stages: a silent stage and a spike stage. The silent stage is utilized to decompose audio information into distinct frequency components and store this data in the state of the RF. The spike stage then oscillates phase information, effectively converting it into ITD cues through phase-locking mechanisms. These stages can be described as follows:

$$\mathcal{Z}_{\mathrm{RF}}[t] = \begin{cases} e^{\mathrm{i}\omega\Delta t}\mathcal{Z}_{\mathrm{RF}}[t-1] + I[t], & \text{Silent Stage}, \\ \lambda e^{\mathrm{i}\omega\Delta t}\mathcal{Z}_{\mathrm{RF}}[t-1], & \text{Spike Stage}. \end{cases} \tag{2}$$

In the silent stage, RF neurons with distinct $\omega$ values selectively respond to specific resonant frequencies. As illustrated in Fig.2(b), when the frequency $\omega_1$ of the audio input $I[t]$ closely matches the RF neuron's resonant frequency $\omega$, a significant increase in its membrane potential occurs. Conversely, misalignment between these frequencies leads to a slower accumulation of membrane potential. This characteristic offers an energy-efficient alternative to the computationally expensive FT operations. The result of the silent stage can be interpreted as analogous to the initial phase information of each pure sinusoidal component within the audio signal.

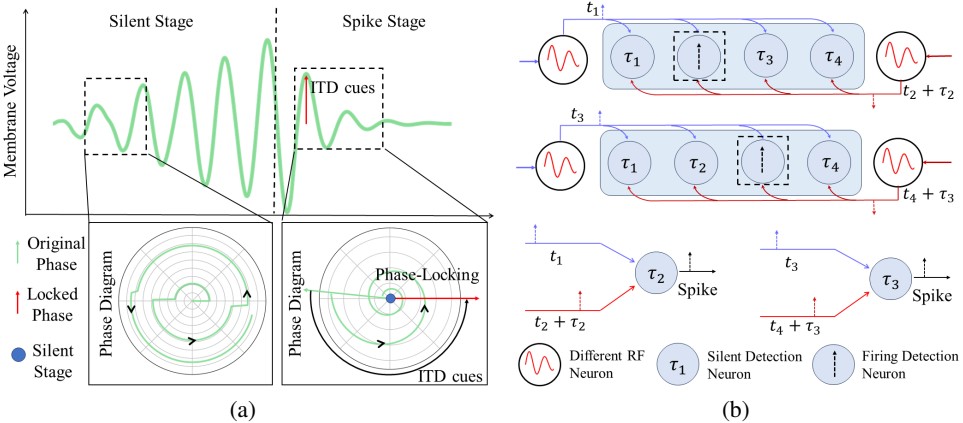

Figure 3: The proposed the RF-PLC method. (a) ITD cues capture: during the silent stage, RF neurons replace FT by responding to input signals. In the spike stage, the RF neurons' first oscillatory peak time is encoded as their spike firing time through a phase-locking loop. (b) The coincidence detection network: detection neurons directly encode ITD cues by analyzing the spike timings of multiple RF neurons from two receivers and generating spikes after a specific time delay.

In the spike stage, we introduce a PLC method that ensures the RF neuron fires a spike only at a specific phase. Specifically, the spike firing time $t_{\mathrm{lock}}$ is defined as the special state when the real part of the RF neuron state reaches zero and the imaginary part reaches its maximum ($\mathcal{Z}_{\mathrm{RF}}[t_{\mathrm{lock}}] = 0 + \mathrm{i}y_{\mathrm{max}}$). This precise spike timing can be directly utilized as an ITD cue, with details validated in Appendix B. Notably, the periodic decay characteristic of RF neurons guarantees that only one spike is generated using our PLC method, ensuring the efficiency of ITD encoding.

As illustrated in Fig. 3(a), we provide a schematic representation for obtaining ITD cues from input audio. During the silent stage, RF neurons receive audio signals and convert them into phase information of pure tones at different frequencies. During the spike stage, the PLC method leverages this phase information to determine whether the RF neuron fires a spike. The precise spike timing of the RF neuron serves as the ITD cue. Compared to traditional FT-based methods that rely on multiple network layers, our RF-PLC significantly reduces computational costs and offers a more biologically plausible representation of ITD cues.

The final step of the RF-PLC method involves detection neurons that convert spike times (also, ITD cues) into spike patterns. As illustrated in Fig. 3(b), a series of detection neurons are used in each band, with each neuron tuned to a specific delay preference (from $\tau_1$ to $\tau_n$). These detection neurons then encode the overall ITD cues for the audio signal. Interestingly, similar symmetrical detection structures have been observed in mammalian auditory pathways [15], which support the biological plausibility of our approach.

## 3.2 MAA: Multi-auditory Attention Mechanism for Back-end SNN-based SSL Model

After encoding audio signals into spike patterns, we construct a back-end SNN-based model to process this encoded information for SSL tasks. The SNN-based SSL model is built based on the Leaky Integrate-and-Fire (LIF) neuron due to its computation efficiency. The LIF model receives the resultant current and accumulates membrane potential which is used to compare with the threshold to determine whether to generate the spike. Its dynamic can be described as follows:

$$U[t + 1] = H[t] + X[t + 1], \tag{3}$$

$$S[t + 1] = \Theta(U[t + 1] - V_{th}), \tag{4}$$

$$H[t + 1] = V_{reset}S[t + 1] + \tau U[t + 1](1 - S[t + 1]). \tag{5}$$

At each time step $t + 1$, the spatial input current $X[t + 1]$ is obtained through convolution and linear layers. This current integrates with the previous temporal input $H[t]$ to update the membrane potential $U[t + 1]$. The Heaviside step function $\Theta(\cdot)$ determines whether the binary spike $S[t + 1]$ is generated by comparing the membrane voltage with the threshold $V_{th}$.

If there is spike emission, $H[t]$ is reset to the resting potential $V_{reset}$; otherwise, $U[t + 1]$ decays with a time constant $\tau$ and directly feeds into $H[t + 1]$. We denote the LIF spiking neuron layer as $\mathcal{SN}(\cdot)$, which takes $X[t + 1]$ as input and produces the spike tensor $S[t + 1]$ as output. Existing back-end SNN-based SSL models only rely on simple convolutional and fully connected layers for localization, without considering biological adaptation mechanisms such as frequency band selectivity and short-term memory. This leaves substantial room for improvement in localization performance. Therefore, we draw on these biological mechanisms to propose the computationally efficient MAA.

In the field of SNNs, there have been some studies on attention mechanisms [19, 65, 71]. However, these methods almost rely heavily on squeeze-and-excitation operations, which introduce additional MAC operations. Therefore, we propose a novel spike-driven MAA mechanism that comprises a frequency-spatial joint attention module and a short-term memory structure. The former enhances networks' focus on critical ITD cues within key frequency bands. The latter strengthens the model's memory for wise decisions across timeframes. Notably, our MAA module is tailored for SSL tasks and achieves the best trade-off between performance and efficiency.

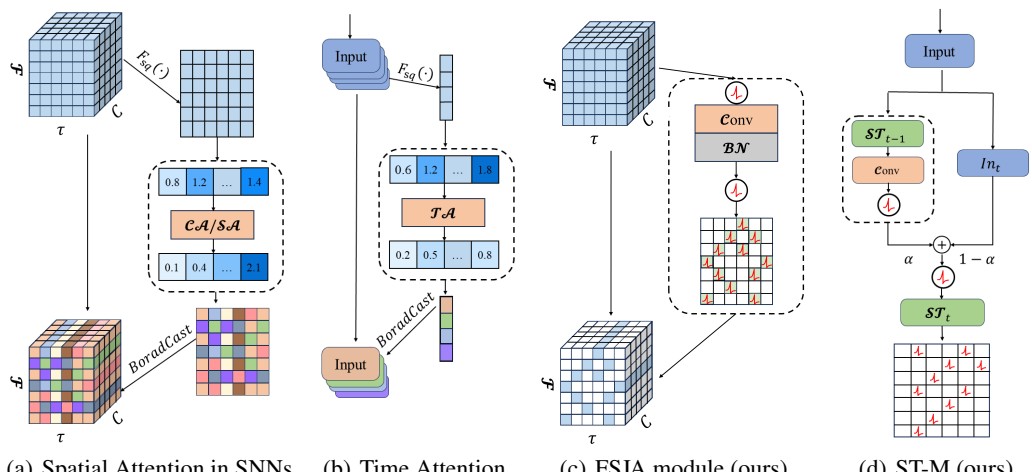

(a) Spatial Attention in SNNs    (b) Time Attention    (c) FSJA module (ours)    (d) ST-M (ours)

Figure 4: Comparing MAA with spiking attention methods. (a) In SNNs, CA/SA[66] uses MAC-based broadcasting operations. (b) TA [65] efficiently focuses on temporal sequences but struggles with streaming data. (c) FSJA adopts a binary attention map as an alternative to MAC-based broadcasting, enhancing computational sparsity and masking noise (white blocks). (d) ST-M incorporates ITD cues within a streaming context, significantly reducing the model's computational resource.

### 3.2.1 Frequency-Spatial Joint Attention

To enhance adaptive learning and selection of preferred frequency in our SNN-based SSL model, we propose a spike-driven FSJA module. For each time step, the output of the RF-PLC method is

defined as $X[t] \in \mathbb{R}^{C \times \mathcal{F} \times \mathcal{S}}$, where $\mathcal{F}$ and $\mathcal{S}$ respectively represent the number of RF and detection neurons, and $C$ denotes the microphone array. The FSJA module can be expressed as follows:

$$\mathcal{Z} = \mathcal{SN}(\frac{1}{\mathcal{F} \times \mathcal{S}} \sum_{i=1}^{\mathcal{F}} \sum_{j=1}^{\mathcal{S}} X_{i,j}[t]), \text{Att}_{FS}(\mathcal{Z}) = \mathcal{SN}(\text{ConvBN}(\mathcal{Z})), \text{FSJA} = \text{Att}_{FS}(\mathcal{Z}) \cdot X[t], \quad (6)$$

where ConvBN is convolution operations with a $3 \times 3$ kernel and batch normalization. The matrix $\mathcal{Z}$ is defined as the average of $X[t]$ across the $\mathcal{F}$ dimension and $\mathcal{S}$ dimension and the spike results after passing through the LIF neuron. Due to the attention map of FSJA module is in binary spike form, it effectively concentrates on spike information at specific frequency and spatial dimensions.

To further demonstrate the difference between our FSJA module and previous SNN-based attention methods across frequency and spatial dimensions, we show the difference between them. As shown in Fig. 4(a), the existing SNN-based attention module relies on full-precision values. Although the broadcast operation is a spike-driven computational paradigm, it introduces additional MAC operations for the next layer. As shown in Fig. 4(c), it effectively avoids MAC-based broadcasting operations which substantially improves energy efficiency. Additionally, our method effectively masks the ITD information of irrelevant frequency bands and spatial regions, substantially boosting the SSL model's robustness in noisy environments.

### 3.2.2 Short-term Memory Structure

The auditory short-term memory characteristic enables sustained perception of SSL processing, yet few studies have focused on this aspect. Although the membrane potential accumulation of spiking neurons partially reflects this mechanism, its simplified mathematical expression is insufficient for describing short-term memory adequately. Therefore, we develop an innovative ST-M structure that emphasizes the interaction of information across adjacent time steps to enhance the neuronal memory capacity. The structure can be represented as:

$$\begin{aligned} In[t] &= \mathcal{SN}(\text{ConvBN}(X[t])), \\ \mathcal{ST}[t] &= \mathcal{SN}(\alpha \, \text{ConvBN}(\mathcal{ST}[t-1]) + (1-\alpha) \, In[t]), \end{aligned} \quad (7)$$

where $In[t]$ represents the preliminary feature extraction of the input $X[t]$, and $\mathcal{ST}[t]$ denotes the enhanced memory unit, with $\alpha$ serving as the hyperparameter that balances adjacent time steps. Our ST-M architecture is asynchronous, processing information frame-by-frame rather than employing time-dimension attention, thereby significantly reducing the computational resources required by the network and facilitating the direct processing of streaming audio information. Additionally, by balancing the memory residual at time $t-1$ with the input information at time $t$, our approach facilitates memory interaction between adjacent time steps. This processing paradigm that relies on input from adjacent time steps pays more attention to short-term memory, thereby granting our model improved localization robustness in dynamic environments.

To avoid the MAC operations present in $\alpha \, \text{ConvBN}(\mathcal{ST}[t-1])$, we integrate $\alpha$ into the firing threshold of $\mathcal{SN}$, which can be expressed by the following formula:

$$\mathcal{ST}[t] = \mathcal{SN}'\left(\text{ConvBN}(\mathcal{ST}[t-1]) + \frac{1-\alpha}{\alpha} In[t]\right). \quad (8)$$

Here, $\mathcal{SN}'$ denotes a layer of spiking neurons with a threshold of $V_{th}/\alpha$. Due to $\mathcal{ST}[t]$ and $X[t]$ being binary spikes and ConvBN can be fused during inference, Eq. 8 contains no MAC operations which ensure low power consumption in inference. Compared with attention in the temporal dimension, the ST-M structure demonstrates asynchronous inference and low-power computational characteristics. As shown in Fig. 4(b) and Fig. 4(d), TA methods require the processing of all temporal information and rely on full-precision attention representation, whereas our ST-M structure utilizes only the spike information from adjacent time steps and features spike-driven computation. This ensures the model can perform inference in a low-power manner.

Combining the FSJA and ST-M modules, we propose a spike-driven MAA mechanism, with its insertion location detailed in the Appendix. D. The proposed MAA mechanism leverages the FSJA module to effectively filter noise, and the ST-M module to strengthen the model's memory for wise decisions across timeframes. As a result, our biologically inspired MAA significantly improves the localization accuracy and robustness of our SNN-based back-end network model.

## 4 Experiments

In this section, we evaluate our proposed spike-based SSL framework performance on three datasets: the HRTF [57], Single Words [30], and SLoClas dataset [42]. Moreover, we examine its energy efficiency and robustness through extensive ablation studies and noise addition experiments on the SLoClas dataset.

### 4.1 Comparison with SOTA Models

The HRTF dataset and Single Word are examined utilizing 2-channel audio at a singular frequency, with a minimum angular resolution of $10°$. In contrast, the SLoClas dataset comprises 4-channel audio in real-world scenarios, with a higher resolution of $5°$. Consequently, the SLoClas dataset presents a higher level of challenge and more closely resembles real-world scenarios. We report in detail the Mean Absolute Error (MAE) and the classification accuracy (Acc.) [7], defined as shows:

$$\text{MAE}(°) = \frac{1}{N} \sum_{i=1}^{N} |\hat{\theta}_i - \theta|,$$

$$\text{Acc.}(\%) = \frac{1}{N} \sum_{i=1}^{N} (|\hat{\theta}_i - \theta| < \eta), \tag{9}$$

where $\hat{\theta}_i$ represents the estimated azimuth angle, and $\theta_i$ denotes the ground truth azimuth angle of the sample i. MAE quantifies the deviation between predicted and true angles, where a lower value means superior performance. Moreover, Acc quantifies the similarity between predicted angles and actual output angles. The $\eta$ is set differently across datasets to align with their specific characteristics: For the HRTF and Single Word datasets, $\eta$ is set at $5°$, while for the SLoClas dataset, it is set to $2.5°$. In this manner, $\eta$ is rounded to the nearest increment corresponding to the minimum localization resolution when calculating classification accuracy.

Table 1: Comparison of sound source localization systems.

| Dataset | Methods | Type | Param (M) | T | DoA | |
|---------|---------|------|-----------|---|------|------|
| | | | | | MAE(°) | Acc(%) |
| HRTF | LSO [57] | SNN | - | - | - | 74.56% |
| | MNTB [57] | SNN | - | - | - | 97.38% |
| | Our works | SNN | 1.64M | 4 | - | 99.84% |
| Single Word | MSO/LSO [30] | SNN | - | - | - | 96.30% |
| | Our works | SNN | 1.64M | 4 | - | 99.63% |
| SLoClas | GCC-PHAT [41] | ANN | 4.17M | - | 4.39° | 86.94% |
| | SELDnet [1] | ANN | 1.68M | - | 1.78° | 88.24% |
| | EINV2 [4] | ANN | 1.63M | - | 0.98° | 94.64% |
| | SRP-DNN [64] | ANN | 1.64M | - | 0.96° | 94.12% |
| | FN-SSL [59] | ANN | 1.68M | - | 0.63° | 95.40% |
| | MTPC-CSNN [40] | SNN | 1.61M | 4 | 1.23° | 93.95% |
| | MTPC-CSNN [40] | SNN | 1.61M | 8 | 1.02° | 94.72% |
| | MTPC-RSNN [40] | SNN | 1.67M | 51 | 1.48° | 94.30% |
| | Hybrid Coding [7] | SNN | 1.61M | 4.37 | 0.60° | 95.61% |
| | Our works | SNN | 1.64M | **4** | **0.33°** $\pm 0.02°$ | **96.40%** $\pm 0.3\%$ |

As shown in Table 1, our model achieves SOTA accuracy among similarly sized models but also significantly reduces MAE metrics. Specifically, our model achieves an accuracy of 99.84% and 99.63% on the HRTF datasets and Single Words, respectively. Additionally, on the challenging SLoClas dataset, our model achieves a MAE of $0.33°$ and an accuracy of approximately 96.4%, while the number of the model parameter is only 1.64M. It represents a nearly 50% improvement in the MAE metric compared to the current SOTA performance of SNN-based models. The localization precision of our model is also competitive compared to other recently introduced ANN models.

## 4.2 Ablation Study

To assess the efficiency of the proposed RF-PLC method and MAA module, we conduct a series of ablation studies. Specifically, we compare our ITD extraction approach in the RF-PLC method with the established FT-based method used in previous work [40]. As depicted in Fig.5(a), our method achieves an accuracy nearly identical to that of the conventional approaches, with an error rate of only 1%. Furthermore, prior research has demonstrated that RF neurons exhibit significantly lower energy consumption compared to FT operations, particularly when implemented on neuromorphic hardware [17, 18, 45]. These findings validate the effectiveness of our RF-PLC method in achieving a highly efficient and accurate ITD encoding scheme.

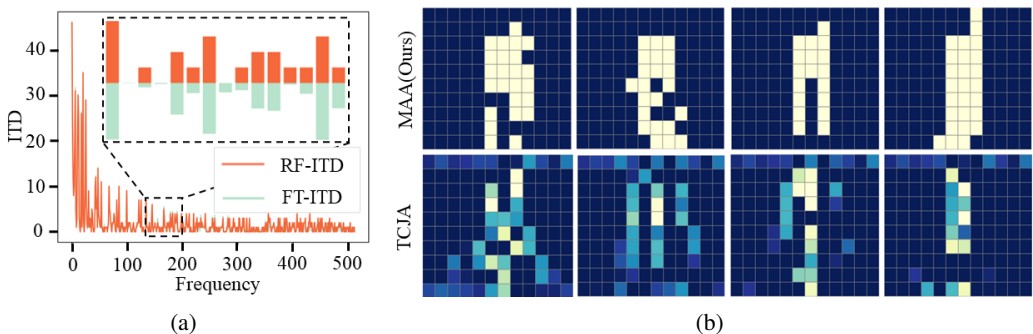

Figure 5: (a) RF-PLF achieves results similar to FT-ITD, highlighting the benefit of avoiding FT operations in ITD encoding. (b) Attention mechanism. MAA's binary attention map effectively filters noise and avoids energy-intensive MAC-based broadcasting compared to other spiking attention.

The effectiveness of the MAA module is demonstrated in the model's localization performance. As shown in Table. 2, both the FSJA module and ST-M structure components individually enhance the performance of the back-end SSL model, and their combination yields even superior results. In addition, compared to attention mechanisms such as TA and TCJA, our attention method employs a fully spike-driven computational paradigm. This characteristic allows our MAA method to maintain an energy consumption of 9.58uJ, representing an increase of 8.49% compared to the work[40]. Moreover, Fig.5(b) illustrates the binary nature of the MAA attention map. This design effectively avoids the energy-intensive broadcasting operations typically associated with MAC units. Details on the energy consumption calculations are provided in Appendix. D and Appendix. E. This capability substantially improves the model's robustness, which is discussed in the following section.

Table 2: Ablation study

| Methods | Spike-Driven | Param (M) | Power (uJ) | DoA | |
| --- | --- | --- | --- | --- | --- |
| | | | | MAE ($°$) | Acc ( % ) |
| Baseline [40] | ✓ | 1.61M | 8.83 | $1.23°$ | 93.95% |
| TA [66] | ✗ | 1.62M | 15.37 | $0.65°\pm0.05°$ | 93.37% $\pm1.2\%$ |
| TCJA [71] | ✗ | 1.68M | 15.34 | $0.47°\pm0.03°$ | 93.45% $\pm1.0\%$ |
| ST-M | ✓ | 1.62M | 8.99 | $0.45°\pm0.03°$ | 95.67% $\pm0.5\%$ |
| FSJA | ✓ | 1.63M | 9.42 | $0.49°\pm0.02°$ | 95.95% $\pm0.6\%$ |
| MAA | ✓ | 1.64M | 9.58 | $\mathbf{0.33°}\pm0.02°$ | $\mathbf{96.40\%}\pm0.3\%$ |

## 4.3 Robustness Experiments

To assess the robustness of our proposed spike-based SSL framework, we evaluate the distribution of MAE under varying signal-to-noise ratio (SNR) conditions. It is a metric used to measure the level of noise present in the input signal. It can be defined as follows:

$$\text{SNR (dB)} = 10 \cdot \log_{10}\left(\frac{P_{\text{signal}}}{P_{\text{noise}}}\right), \tag{10}$$

where $P_{\text{signal}}$ represents the power of the signal and $P_{\text{noise}}$ denotes the power of the noise. A lower SNR indicates a higher proportion of noise. Specifically, we incorporate noise from the NOISEX-92 database into audio recordings from different microphone channels. Detailed information about the noise addition process and the experimental setup is described in the Appendix.C.

As shown in Fig.6(a), we visualize the encoding results of the RF-PLC method under various SNR conditions to better understand the input form of our back-end model. In addition, we also present the distribution of recognition over $360°$. As shown in Fig.6(b) and Fig.6(c), the MTPC method is more likely to predict certain angles, especially in terms of information in the noise direction; however, our method is not significantly affected by noise information. It indicates that our model exhibits higher stability. As SNR increases, our method demonstrates higher recognition accuracy. This indicates that our model effectively suppresses noise in specific frequency bands, thereby preventing significant variations in recognition results due to increased noise. The results provide strong evidence of the model's superior generalization and robustness when applied to complex real-world scenarios.

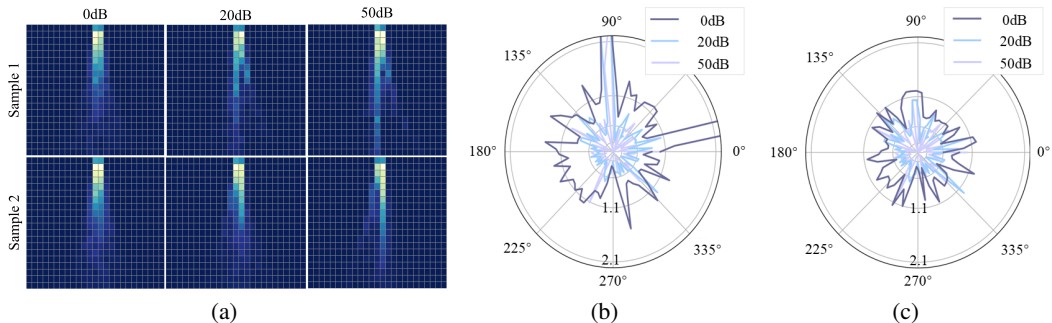

Figure 6: Performance under varying SNR levels. (a) Impact of SNR on ITD Encoding: at 0 dB, it is challenging to intuitively discern the direction of the sound source from the encoding results. (b) and (c) Different distribution of MAE over $360°$ in MTPC [40] and our model. Our model achieves enhanced noise resistance and improved localization stability.

## 5   Conclusion

Inspired by the efficiency of biological auditory systems, this work proposes a novel spike-based SSL framework. The core components are the RF-PLC method and the MAA module. The RF-PLC method leverages the resonance properties of RF neurons to bypass computationally expensive FT operations. It utilizes a phase-locking loop and ITD detection neurons to efficiently encode ITD cues from the audio signal into spike trains. Furthermore, the study incorporates insights from auditory biology, including frequency preferences and short-term memory characteristics. By designing a fully spike-driven MAA module, our SNN-based SSL model effectively filters irrelevant environmental noise in the frequency domain while temporally focusing on specific auditory content. This approach achieves superior performance, robustness, and interpretability, significantly advancing the field of neuromorphic SSL research. It establishes a new benchmark for the development of SSL techniques. Future work will investigate the deployment of this model on neuromorphic hardware platforms.

## 6   Acknowledgments

This work was supported in part by the National Natural Science Foundation of China under grant U20B2063, 62220106008, and 62106038, the Sichuan Science and Technology Program under Grant 2024NSFTD0034 and 2023YFG0259.

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

## A    RF Neurons for Energy Efficient FT alternative

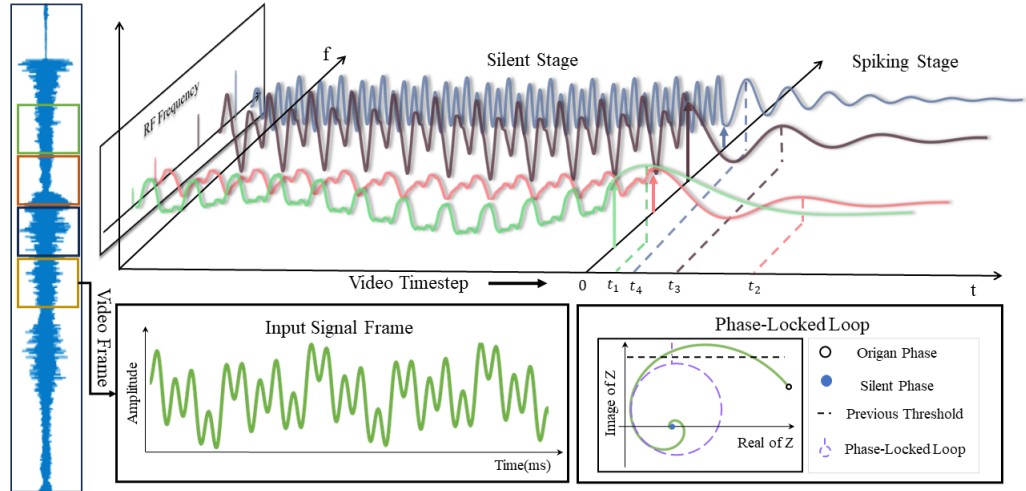

Figure 7: RF neurons serve as an energy-efficient alternative to FT, accumulating membrane potential during silent phases to substitute for FT, and directly mapping phases during spike phases.

**Lemma 1. Nyquist Theorem**
The Nyquist Theorem is pivotal for the sampling process in converting analog signals to digital signals. It stipulates that to avoid aliasing, the sampling frequency must be at least twice the maximum frequency component present in the analog signal. This criterion ensures that the reconstructed digital signal closely approximates the original analog signal without distortion.

**Lemma 2. Fourier Transform (FT)**
Consider an audio sequence $x = [x_1, x_2, \ldots, x_T]$ sampled at frequency $f_s$. The Fourier Transform (FT) facilitates the conversion from time domain to frequency domain, computed as:

$$\mathcal{F}[k] = \sum_{n=0}^{N-1} x[n] e^{-\mathrm{i}\frac{2\pi}{N}nk} = \sum_{n=0}^{N-1} x[n] \left( \cos\left(\frac{2\pi}{N}nk\right) - \mathrm{i}\sin\left(\frac{2\pi}{N}nk\right) \right), \tag{11}$$

where $N$ is the number of discrete samples used in the FT. The complex vector $\mathcal{F}[k]$ quantifies the spectral components at varying frequencies. Utilizing Lemma 1, these components represent sinusoidal signals decomposed at frequencies indexed by $k$, scaled by $\frac{f_s}{N}$. For each component $\mathcal{F}[k] = a_k + ib_k$, the corresponding time-domain signal can be described by:

$$y_k(t) = \sqrt{a_k^2 + b_k^2} \sin\left( 2\pi \frac{f_s}{N} kt + \tan^{-1}\left(\frac{b_k}{a_k}\right) \right). \tag{12}$$

**Proof:** Assume a series of RF neurons, each with a resonant frequency of $\omega = [-0 * \frac{2\pi}{N}, -1 * \frac{2\pi}{N}, -2 * \frac{2\pi}{N}, \ldots, -(N-1) * \frac{2\pi}{N}]$, initially set to zero. When exposed to a real-time input audio $x$, the response of the $k^{th}$ neuron at time $t$ is given by the recursive update:

$$\begin{aligned}
Z_{\mathrm{RF}_k}[t] &= x[t] + \lambda e^{\mathrm{i}\omega_k \Delta t} Z_{\mathrm{RF}_k}[t-1] \\
&= x[t] + \lambda e^{\mathrm{i}\omega_k \Delta t}(x[t-1] + \lambda e^{\mathrm{i}\omega_k \Delta t} Z_{\mathrm{RF}_k}[t-2]) \\
&= \sum_{n=1}^{T} \lambda^n e^{\mathrm{i}n\omega_k \Delta t} x[t-n] = \sum_{n=1}^{T} \lambda^n \left( \cos\left(n\omega_k \Delta t\right) - \mathrm{i}\sin\left(n\omega_k \Delta t\right) \right) x[t-n].
\end{aligned} \tag{13}$$

This recursive filtering mimics a discrete Fourier transform when $\lambda = 1$. Moreover, RF neurons can be efficiently implemented on neuromorphic hardware like the Loihi2 chip, facilitating low-power and high-speed computations.

## B   Initial Oscillation Peak of RF Neurons as ITD Cue

**Lemma 1. Neural Phase Coder**
Inspired by observations of specific mechanisms within the auditory and visual cortices [9, 26], biological evidence are expressed through phase coding found. It can encode the pure tone audio into precise-timing spike. Regarding the results of the decomposition:

$$y_L = A_1 \sin\left(2\pi f_1 t + \tan^{-1}\left(\frac{b_{k_L}}{a_{k_L}}\right)\right), y_R = A_2 \sin\left(2\pi f_1 t + \tan^{-1}\left(\frac{b_{k_R}}{a_{k_R}}\right)\right), \qquad (14)$$

$y_L$ and $y_R$ represent the single-tone sinusoidal signals arriving at the left and right ears. Subsequently, the first peak time is encoded into spike time as the arrival time of the sound.

$$t_L = \frac{1}{2\pi f}\left(\frac{\pi}{2} - \tan^{-1}\left(\frac{b_{k_L}}{a_{k_L}}\right)\right), t_R = \frac{1}{2\pi f}\left(\frac{\pi}{2} - \tan^{-1}\left(\frac{b_{k_R}}{a_{k_R}}\right)\right), \qquad (15)$$

where $f$ represents the frequency of the sinusoidal signal. Thus, ITD can be expressed as $t_L - t_R$.
**Proof:** When RF neurons (from both ears) enter the spike stage, their initial states are represented as $a_{k_L} + ib_{k_L}$ for the left ear and $a_{k_R} + ib_{k_R}$ for the right ear. Here, $L$ indicates the left ear, $R$ denotes the right ear, and $K$ refers to the RF neuron associated with the intrinsic frequency $f_k$. Subsequently, the state will decay oscillations over time. To facilitate understanding, we calculate the real and imaginary parts in a discrete manner:

$$\begin{cases} a_{k_L}[t] = a_{k_L}[t-1]\cos(2\pi f_k) + b_{k_L}[t-1]\sin(2\pi f_k), \\ \\ b_{k_L}[t] = -a_{k_L}[t-1]\sin(2\pi f_k) + b_{k_L}[t-1]\cos(2\pi f_k), \end{cases} \qquad (16)$$

With the RF-PLC method we propose, we can directly spike timing acquisition. Specifically, RF neurons will fire spike at $a_{k_L}[t_{n_L}] = 0, b_{k_L}[t_{n_L}] = \max(b_{k_L})$. In the phase space, this state represented the first peak of time aligns with the neural phase coder:

$$\begin{cases} \phi_{locked} = \tan^{-1}\left(\frac{b_{k_L}[t_{n_L}]}{a_{k_L}[t_{n_L}]}\right) = \frac{\pi}{2}, & \text{Phase-locking Loop} \\ \\ \text{ITD}_{RF} = t_{n_L} - t_{n_R} \approx t_L - t_R, & \text{RF-based ITD encoding} \end{cases} \qquad (17)$$

where $t_{n_L}$ satisfies $\phi_{locked} = \mathcal{Z}_{RF}[t_{n_L}]$ and $t_{n_R}$ satisfies $\phi_{locked} = \mathcal{Z}_{RF}[t_{n_R}]$. Due to the discrete form, our ITD encodings are not identical to those in Lemma 1. The error is in the difference between the audio's sampling rate and the actual frequency. It results in our inability to accurately obtain the first peak time.

Specifically, as Lemma 1 demonstrates, to obtain the spike timing for pure tone audio of different frequencies, the phase coding model requires leveraging Eq.15 to compute the audio's ITD cues. However, with the RF-PLC method we propose, it can only iterate according to the audio's sampling rate. The difference between them is dependent on the sampling rate $f_s$, which can be represented as $1/f_s$. For the dataset SLoClas that we tested, its errors is only approximately 1%.

## C   Experiment Detail

We primarily validated the accuracy and robustness of our proposed method on the SLoClas dataset. It utilizes a 4-channel microphone array to collect data on RWCP sound scenes to ensure SNR of 40 to 50dB. It is comprised of ten distinct categories of ambient sounds: bells, bottles, buzzers, cymbals, horns, metal, particles, phones, rings, and whistles. Each category includes approximately 100 instances, providing a diverse and comprehensive set of audio samples.

Additional, to evaluate the robustness of our method, we need to construct source localization information under various complex scenarios. Specifically, we can introduce different types of noise into the audio. The noise sounds are sourced from the NOISEX-92 database, which contains recordings of various types of real-world noise. We add noise audio to each microphone channel to simulate noise coming from different directions.

$$ComplexVideo_i(n) = mic_i(n) + \lambda noise(n), i = 1, 2, 3, 4. \tag{18}$$

In this setup, $mic_i(n)$, where $i = 1, 2, 3, 4$, represents the 4-channel signals recorded by the microphone array. ComplexVideo$_i$ denotes the multi-channel data with noise. The noise data consists of randomly selected audio clips from a noise database. $\lambda$ is the scaling factor used to adjust the audio to a specific SNR ratio. A smaller SNR indicates a stronger noise presence in the audio which is more similar to real environment.

Table 3: Experimental configuration of the sound localization task.

| Attributes | Setup |
|---|---|
| **1. Data preprocessing:** | |
| Sampling rate (Hz) | 16000 |
| Frame length (ms) | 170 |
| Frame stride (ms) | 170 |
| RF neurons $n$ | 512 |
| Number of Microphones | 4 |
| **2. RF-PLC setting:** | |
| CQT frequency range (Hz) | [0, 8800] |
| $\tau$ (ms) | 0.0625 |
| Frequency channels $N$ | 40 |
| Coincidence detector $N_\tau$ | 51 |
| Microphone pairs $C$ | 6 |
| **3. SNN Hyperparameter:** | |
| $\alpha$ | 0.75 |
| Timestep | 4 |
| Epochs | 300 |
| Batch size | 128 |
| Optimizer | Adam |
| Base learning rate | 1e-3 |
| Learning rate decay | Cosine |
| Weight decay | 5e-3 |

## D  Model Structure

The overall network architecture of our SNN-based SSL model is illustrated in Fig.8, featuring a comprehensive system design tailored for sound localization. The architecture consists of two main components: a front-end RF-PLC method and a back-end MAA-based localization decision network.

In the RF-PLC method, we utilize 512 RF neurons with widths $\omega$ that increase incrementally from 0 to 8000Hz. These neurons are strategically deployed as an alternative to the traditional FT operation, optimizing the model for energy efficiency and computational speed. Additionally, we utilize a cochlear filter bank following the Constant Q Transform (CQT) which is the most commonly used and easily implemented cochlear filter bank, to extract auditory features of appropriate dimensions. This encoding approach not only mimics the cochlear filtering process but also enhances the temporal dynamics of sound processing. Additionally, $N$ detection neurons are engaged to characterize the ITD with delays ranging from -25$\tau$ to 25$\tau$. Based on the audio sample ratio, the value of $\tau$ is set at 0.0625 ms. Furthermore, our model utilizes data from four microphones to compute ITDs between each pair, resulting in six distinct sets of ITD cues. As a result, each 170 ms speech frame is encoded into $X \in \mathbb{Z}^{C \times N \times N_\tau}$, capturing a rich array of spatial and temporal information. Details of these parameters can be found in Table.3. This setup ensures a detailed and dynamic spatial representation of auditory scenes.

In the back-end decision network, we propose a fully spike-based model, which is illustrated in Fig. 8. We will validate our module within the SSL models [7, 40] consisting of convolutional and MLP layers. To enhance the performance of the SSL model, we augment this basic structure with our MAA

module. This module is designed to emulate biological auditory processing by focusing on frequency band preference and short-term memory capabilities. Compared with traditional spatio-temporal spike attention techniques [65, 71], our module markedly improves the system's computational efficiency, as shown in Table. 2. Additionally, to better illustrate the performance of our module, we present a comparison with various modules. Details of these networks are provided in Table. 4. This enhancement allows for faster and more accurate sound localization, demonstrating the potential of our model in real-world applications.

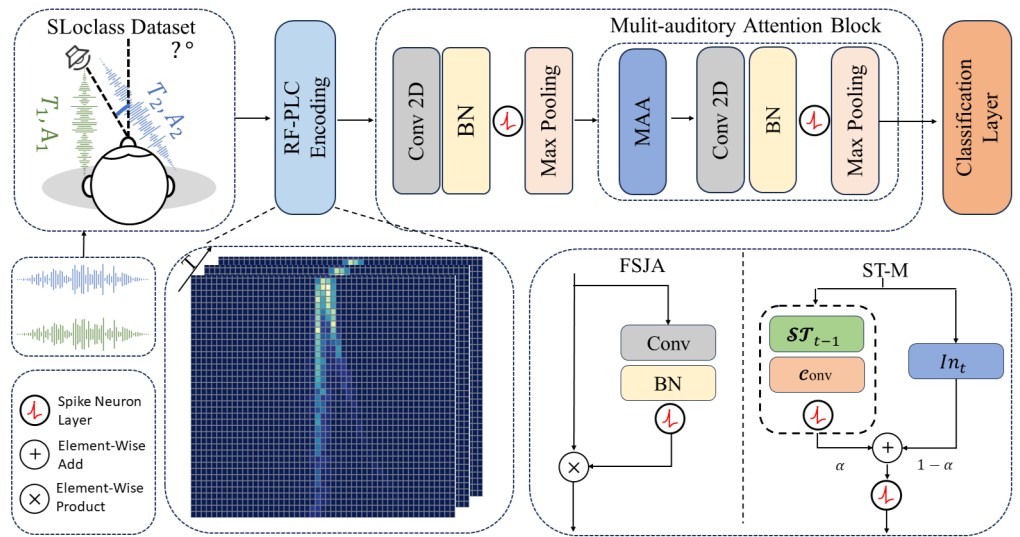

Figure 8: The structure of the Spike-based Neuromorphic Sound Source Localization. It includes an RF-PLC encoding method and a back-end classification model based on SNNs.

Table 4: Detail Network

| Stage | Output Size | Baseline | Baseline + MAA | Baseline + others |
|-------|-------------|----------|----------------|-------------------|
| Stage 1 | $12 \times 25 \times 20$ | Conv $3 \times 3$, stride 1
BatchNorm
MaxPooling $2 \times 2$, stride 2 | | |
| Stage 2 | $24 \times 12 \times 10$ | Conv $3 \times 3$, stride 1
BatchNorm
MaxPooling $2 \times 2$,stride 2 | MAA module | other Attention |
| Stage 3 | $48 \times 6 \times 5$ | Conv $3 \times 3$, stride 1
BatchNorm
MaxPooling $2 \times 2$,stride 2 | MAA module | other Attention |
| Classifier | $1 \times 1 \times 1$ | 360-FC | | |

# E    Energy Cost

To describe the energy consumption calculations in the ablation experiments, we introduce a theoretical energy estimation method for the proposed attention mechanism. Compared to the ANN model, the energy consumption calculation of the spiking version requires information on the timesteps (T) and spike firing rates (R). The spike firing rate is defined as the proportion of non-zero elements in the spike tensor. Since our proposed MAA method is spike-driven, we only need to evaluate the model's FLOPs, along with T and R, to estimate the theoretical energy consumption of our methods.

In ANN [35], the FLOPs for the n-th Conv layer are expressed as:

$$\mathcal{C}onv = (k_n)^2 \cdot h_n \cdot w_n \cdot c_{n-1} \cdot c_n, \tag{19}$$

where $k_n$ denotes the kernel size, $(h_n, w_n)$ specifies the dimensions of the output feature map, and $c_{n-1}$ and $c_n$ represent the numbers of input and output channels, respectively. The FLOPs of the m-th MLP layer in ANNs are:

$$\mathcal{MLP} = i_m \cdot o_m, \tag{20}$$

where $i_m$ and $o_m$ represent the input and output dimensions of the m-th MLP layer.

Refer on previous research [24, 68], we assume that all computational data are implemented using 45nm technology for 32-bit floating-point calculations, with $E_{MAC} = 4.6$pJ and $E_{AC} = 0.9$ pJ.

In Table 3, we further present the details of different models. Referring to [66, 71], the attention matrix is derived using the sigmoid function, which results in the network input for the subsequent layer being non-spiking. Consequently, during the energy consumption calculation, this component is computed using the energy consumption of MAC operations, leading to a significant increase in energy consumption within the network. Due to the unique properties of the MAA attention matrix, our method does not introduce additional floating-point operations.

## F  Limitation

The limitations of this study include the lack of deployment on edge devices. Furthermore, the limited availability of datasets for SSL tasks restricts the validation of our model across a broader array of datasets. Future research will seek to overcome these challenges by amassing a more extensive collection of datasets and implementing our model on edge devices to more effectively ascertain the efficacy of our approach. The experimental results reported herein are reproducible, with detailed descriptions of the model architectures and hyperparameter configurations available in Appendix. Additionally, our code will be made available on subsequent after review.

## G  Supplementary Ablation Experiments

To further validate that the enhancement in our model's performance is indeed due to the effective implementation of band selection and short-term attention mechanisms, rather than an increased parameter count. We designed a Conv2d layer module with the same amount of parameters, ensuring all other parameters remained consistent. Specially,

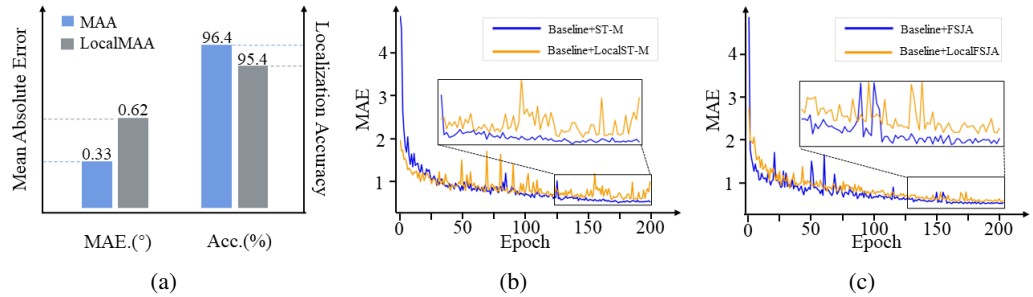

Figure 9: Supplemental ablation experiments. In these experiments, the Local module is configured with the same number of parameters as our proposed models to ensure a fair comparison: (a) A comparison of MAE.(°) and Acc.(%) between our proposed MAA and LocalMAA. (b) A comparison of the MAE between our proposed ST-M and LocalST-M. (c) A comparison of the MAE between FSJA and LocalFSJA.

As illustrated in Fig. 9, LocalMAA refers to a network with the same number of parameters as our proposed MAA. It can be observed that our MAA method achieves a lower MAE(°) and higher Acc(%). Furthermore, we compared the MAE of different modules, finding that our frequency band preference and short-term memory structure significantly enhance the network's localization performance.

