# OpenReview forum: "Spike-based Neuromorphic Model for Sound Source Localization"
_NeurIPS.cc/2024/Conference — NeurIPS 2024 poster_

### Official Review · Reviewer_uJwn · 2024-07-10

**Soundness:** 3
**Presentation:** 3
**Contribution:** 3
**Rating:** 7
**Confidence:** 5

**Summary:**

This study draws inspiration from the intrinsic mechanisms of sound source localization in biological auditory systems to design an efficient and robust SSL model. The core contributions include two primary aspects: firstly, replacing the energy-intensive Fourier Transform(FT) operations with RF neurons in the auditory encoding process, thereby encoding Interaural Time Differences (ITD) as spatiotemporal spike trains. Secondly, the backend SSL model effectively implements frequency band preferences and memory-based decision-making processes observed in biological hearing.

**Strengths:**

Overall, this is an interesting work. The strengths are outlined as follows:

1. The combination of RF neurons and Jeffress models creates a robust auditory localization encoding method. RF neurons effectively replace high-energy Fourier Transform operations. Additionally, the multi-band cooperative decision-making for ITD encoding significantly enhances robustness.
2. The authors have thoroughly considered the full-spike information transmission mechanism of SNNs, providing a composite spike-driven attention paradigm for similar future tasks. Figure.5 clearly illustrates the advantages of the proposed spiking attention.
3. The article provides a clear description of the biological plausibility of the encoding and backend SSL model, making it easy to understand.
4. The visualizations within the paper are outstanding. The diagrams in Figures 1 and 4 are densely packed with information, yet the text and captions effectively guide the reader to comprehend them well.
5. The mathematical proofs in the appendix are comprehensive and credible.

**Weaknesses:**

While the paper is generally convincing, I believe the following areas require attention:
1. The claim that RF neurons can effectively replace high-energy Fourier Transform operations needs further experimental validation or existing research evidence. I suggest the authors bolster this section with reliable proof.
2. While there are currently no datasets to test localization robustness, the experiments on robustness lack detailed descriptions of how noise was introduced in the SLoClas dataset.
3. Despite the high cost of deploying the proposed sound localization model on hardware, I recommend that the authors attempt actual deployment on neuromorphic agents in future work to empirically verify model performance.

**Questions:**

1. The short-term memory structure proposed seems to only consider relationships between adjacent frames. Could methods involving multi-frame integrated decision-making be explored?
2. The authors point that RF neurons can effectively perform the basilar membrane's frequency band decomposition in the cochlea. Is there a biological precedent for similar RF neurons in the cochlea?

**Limitations:**

While the proposed neuromorphic SSL model demonstrates promising results, several limitations should be acknowledged:

1. **Lack of Edge Device Deployment:** The current study does not include the deployment of the proposed SSL model on edge devices. Practical implementation and testing on such devices are crucial to validate the model's real-world applicability and performance.

2. **Experimental Validation of RF Neurons:** The claim that RF neurons can replace energy-intensive Fourier Transform operations lacks sufficient experimental validation. Further empirical evidence or references to existing research are needed to substantiate this claim convincingly.


By addressing these limitations, future research can provide a more comprehensive evaluation of the neuromorphic SSL model and its potential impact on sound source localization and broader applications in neuromorphic computing.

---

> ### Author Rebuttal · Authors · 2024-08-06
>
> Thank you very much for your appreciation of our paper. In response to the issues you raised, we provide the following responses:
>
> # Q1: Mathematical Equivalence of RF Neurons and FT Operations
>
> **A**: The mathematical proof demonstrating the **equivalence** of RF neurons as an efficient substitute for FT is provided in Appendix A, with the equivalence relationship expressed as follows:
>
> $ FT[k] = \sum_{n=0}^{N-1} x[n] e^{-\mathrm{i}\frac{2\pi}{N}nk} = \sum_{n=0}^{N-1} x[n] \left( \cos{\left(\frac{2\pi}{N}nk\right)} - \mathrm{i} \sin{\left(\frac{2\pi}{N}nk\right)} \right) $
>
> $ RF[k] = \sum_{n=1}^{T} \lambda_n \cos(n\omega_k \Delta t) - \mathrm{i} \sin(n\omega_k \Delta t) x[t-n] $
>
> Moreover, In Figure 5(a), we validated the error between RF-ITD and FT-ITD across a substantial audio dataset, and the **error ratio** between these two methods was only **1%**.
>
> # Q2: Why no Datasets for Robustness
>
> **A**: Currently, there are no existing SSL datasets specifically designed for **noisy environments**. To validate robustness in all current SSL tasks, noise is artificially added, as demonstrated in the study [1,2,3]. Furthermore, we have detailed the process of adding noise to the SLoClas dataset and the preparation of robustness experiments in Appendix C.
>
> $ComplexVideo_i (n) = mic_i (n) + \lambda noise(n), i=1,2,3,4$
>
> Reference:
>
> [1] Multi-tone phase coding of interaural time difference for sound source localization with spiking neural networks. In: TASLP (2021).
>
> [2] A hybrid neural coding approach for pattern recognition with spiking neural networks. In: TPAMI (2023).
>
> [3] SLoclas: A database for joint sound localization and classification. In: O-COCOSDA (2023).
>
> # Q3: Deploying the Proposed SSL Model on Hardware
>
> **A**: We fully agree with you. However, we faced significant challenges in this area. Firstly, open-source edge devices, such as robots and robotic dogs, are costly and have inconsistent interfaces. Secondly, the optimal deployment of SNNs is contingent on neuromorphic chips or specialized hardware, which are both scarce and difficult to procure.
>
> # Q4: Further Analysis of Short-term Memory Structures
>
> **A**: We fully agree with your perspective and have conducted extensive experimental studies in ablation study. Our research explored the performance of **adjacent frames (our)**, **multi-frame**, and **global frames (TA [1])**. The ablation study results, shown in the below table, indicate that adjacent frames yield the best performance. This is because, while global and multi-step frames introduce additional useful information, they also introduce **more noise**, hindering the model's ability to quickly capture key ITD features. Moreover, the temporal interactions in global and multi-step frames require multiple audio frames for decision-making, undermining the model's real-time localization.
>
> | model | Type | Param (M) | MAE ($\degree$) | ACC. (\%) |
> |:-----:|:---------:|:----------:|:---------:|:---------:|
> | baseline | SNN | 1.61M | $1.23\degree$ |  $93.95$\% |
> | global frames (TA [1]) | SNN | 1.62M | $0.65\degree \pm 0.05\degree$ |  $93.37$\% $\pm$ $1.2$\% |
> | multi-frames | SNN | 1.62M | $0.62\degree \pm 0.04\degree$  |  $93.47$\% $\pm$ $1.1 $\% |
> | **adjacent frames (ours)** | SNN | 1.62M | **$0.45\degree \pm 0.03\degree$** |  **$95.67$\% $\pm $$0.5$\%** |
>
> Reference:
>
> [1] Temporal-wise Attention Spiking Neural Networks for Event Streams Classification. In: ICCV (2021).
>
> # Q5: A Biological Precedent for Similar RF Neurons in the Cochlea
>
> **A**: Yes, RF neurons do have biological precedents in the cochlea. However, it's important to note that RF neurons are not a simplification of any specific biological cell. Instead, they are a mathematical abstraction of the combined functions of the cochlear basilar membrane and inner hair cells [1,2].
>
> Reference:
>
> [1] Dendritic channelopathies contribute to neocortical and sensory hyperexcitability in Fmr1−/y mice. In: NAT NEUROSCI (2014).
>
> [2] An oscillator model better predicts cortical entrainment to music. In PANS (2019).

---

> ### Comment · Reviewer_uJwn · 2024-08-12
>
> Thank you for the response, which has addressed my concerns very well, so I will increase the score.

---

> > ### Author Response · Authors · 2024-08-12
> > **Thank you for your valuable suggestions and positive feedback.**
> >
> > We are pleased to know that we have addressed your concerns regarding this manuscript. Thank you again for your valuable suggestions to improve the quality of our work.

---

### Official Review · Reviewer_ym4t · 2024-07-10

**Soundness:** 3
**Presentation:** 3
**Contribution:** 3
**Rating:** 8
**Confidence:** 5

**Summary:**

This paper constructs a sound source localization model by leveraging efficient spiking neural networks and biologically-inspired auditory localization mechanisms. Although numerous studies have previously approached this subject from a biomimetic perspective, this paper commendably considers the balance between biomimicry and practicality, while also addressing the robustness of the SSL model. It offers a new perspective on the effective integration of artificial neural networks and biological intelligence.

**Strengths:**

1.The paper proposes substituting the energy-intensive Fourier Transform (FT) operations with frequency band preferences of RF neurons, which is highly suitable for the edge computing scenarios mentioned by the authors. This consideration is notably absent in most related works.
2.In the appendix, the authors provide a detailed and reliable mathematical proof explaining why RF neurons can effectively replace FT operations. The results are robust and convincingly presented.
3.Inspired by the mechanisms of biological auditory localization, the paper designs an efficient SSL encoding and model. This innovative approach is commendable and encourages further exploration in the field.

**Weaknesses:**

1.It remains insufficient for readers without a background in SSL tasks or RF technology. Although the supplementary materials are helpful, they seem not to adequately bridge the knowledge gap for those outside the specialty.
2.I understand that the outputs of the RF-PLC method will serve as inputs for the classification model. It would be beneficial if the authors could provide a step-by-step example that shows how a given audio input is encoded into a spike pattern, and how these encoded outputs are subsequently utilized to determine the azimuth of a sound source relative to the listener.

**Questions:**

1.The authors are asked to explain how to determine the azimuth of a sound source, including an example of encoding an audio segment and predicting the azimuth.
2.The effectiveness of the MAA Block needs verification. While Table 2 shows improved localization performance without added computational demand, further experiments should confirm these results are not due to other factors like increased parameter counts.
3.Although this work focuses on the algorithm of SSL and I agree that the workload is sufficient, I wonder whether the RF neuron has the potential to be implemented on current neuromorphic hardware or FPGA.

---

> ### Author Rebuttal · Authors · 2024-08-06
>
> Thank you very much for your recognition of our work. In response to the questions you raised, we provide the following replies:
>
> # W1: Enhance the Background Introduction
> We strongly agree with your perspective. In the background section, we will incorporate additional research on SSL tasks and compare methodologies. Additionally, we will further augment the appendix with theoretical proofs of the formulas.
> # W2: Intuitive Demonstration of Model
> We greatly appreciate your suggestions. We will revise Fig.1 to more clearly demonstrate the function of each part of the model.
>
> # Q1: How to Determine the Azimuth of the Sound Source
>
> **A**: We will detail the function of each part of the model and its data format, as well as how to ultimately determine the azimuth.
> - **RF-PLC Method**: An audio is initially segmented into a sequence of frames, denoted as $x\in R^{T \times C_{mic} \times L}$, where $T$ represents the number of frame sequences, $C_{mic}$ indicates the number of microphone channels, and $L$ denotes the length of each frame sequence. Subsequently, audio from different channels is paired and input into the RF-PLC encoding section, resulting in the encoding $E\in Z^{T \times C \times F \times \tau}$, where $C$ represents the number of channel combinations, $F$ denotes the number of frequency band decompositions, and $\tau$ indicates the number of detection neurons.
> - **Back-end Model**: Encoding result $E\in Z^{T \times C \times F \times \tau}$ will be input into a network featuring the MAA module. It primarily consists of **frequency band preference** and **short-term memory components**. The frequency band preference ensures the network focuses more on crucial frequency band information, while the short-term memory enhances the temporal capabilities of neurons. Combined, these features significantly boost the environmental robustness of the backend network. Finally，the output layer of the network consists of 360 units, representing the predicted outcomes.
>
> # Q2: Effectiveness of the MAA Module
>
> **A**: To ensure that the effectiveness of the MAA module is **NOT** merely due to an increase in parameter count, we conducted additional ablation experiments detailed in the supplementary materials. As depicted in Fig.9 and below Table, 'Local-XXX' represents the scenario where the module is replaced with a convolutional layer of equivalent parameter size. The results clearly demonstrate that even with an equal number of parameters, our module exhibits superior performance, underscoring its unique benefits.
>
>
> | model | Type | Param (M) | MAE ($\degree$) | ACC. (\%) |
> |:-----:|:---------:|:----------:|:---------:|:---------:|
> |  LocalST-M | SNN | 1.62M | $0.48\degree$ |  $95.40$\% |
> |  ST-M | SNN | 1.62M | $0.45\degree$ |  $95.67$\% |
> |  LocalFSJA | SNN | 1.63M | $0.62\degree$ |  $95.40$\% |
> |  FSJA | SNN | 1.63M | $0.49\degree$  |  $95.95$\% |
> |  LocalMAA ( LocalST-M + LocalFSJA) | SNN | 1.64M | $0.62\degree$ |  $95.40$\% |
> |  **MAA ( ST-M + FSJA)** | SNN | 1.64M | **$0.33\degree$** |  **$96.40$\%** |
>
>
> # Q3: Hardware-friendly of RF Neurons
>
> **A**: We agree with your view. RF neurons and our SSL model are indeed compatible with current neuromorphic platforms like Loihi 2 [1] and Tianjic [2]. Yet, substantial challenges hinder their broader adoption. Primarily, the high costs and varied interfaces of open-source edge devices like robots limit practical deployments. Furthermore, the scarcity and procurement difficulties of neuromorphic chips or specialized hardware impede the effective utilization of SNNs.
>
> Reference:
>
> [1] Efficient Video and Audio Processing with Loihi 2. In: ICASSP (2024).
>
> [2] Towards artificial general intelligence with hybrid Tianjic chip architecture. In: Nature (2019).

---

> > ### Comment · Reviewer_ym4t · 2024-08-12
> > **Good rebuttal**
> >
> > The experimental examples provided have effectively addressed my questions, and the ablation studies on the MAA block also confirm the model's effectiveness. However, I am particularly interested in how the model performs when deployed on neuromorphic chips or FPGAs. From the authors' responses, it appears that the model's inference relies mainly on two parts, which seems to contradict the original intent of designing a system. From a hardware deployment perspective, it is crucial to determine whether the model can function as a complete system for inference, without considering resource scarcity and procurement difficulties.
> >
> > Overall, most of the concerns have been addressed. Thank you for your response. I would like to increase my score to 8.

---

> > > ### Author Response · Authors · 2024-08-12
> > > **Thank you for your  valuable suggestions and positve feedback.**
> > >
> > > Thank you for your response and recognition. We are pleased to know that most of your concerns have been addressed. Your comments are crucial in enhancing the quality of our work.

---

> ### Author Response · Authors · 2024-08-12
>
> Thank you for acknowledging our work. As you per, traditional ITD encoding often relies on FT operations [1], which are difficult to deploy on neuromorphic hardware, resulting in a divided implementation. In contrast, our ITD encoding scheme employs RF neurons, which have been successfully deployed on Loihi 2 [2] with excellent performance. This demonstrates that our encoding method does not face technical barriers when deployed on neuromorphic chips. Additionally, our backend localization model is strictly spike-driven, with extensive research confirming that such models can be effectively implemented on neuromorphic hardware [3, 4]. Thus, our SSL model can be deployed as a complete system on neuromorphic hardware.
>
> [1] A hybrid neural coding approach for pattern recognition with spiking neural networks. In: TPAMI (2024).
>
> [2] Efficient Video and Audio Processing with Loihi 2. In: ICASSP (2024).
>
> [3] 22.6 anp-i: A 28nm 1.5 pj/sop asynchronous spiking neural network processor enabling sub-o. 1 μj/sample on-chip learning for edge-ai applications. In: ISSCC (2023).
>
> [4] Towards artificial general intelligence with hybrid Tianjic chip architecture. In: Nature (2019).

---

### Official Review · Reviewer_zZvi · 2024-07-11

**Soundness:** 3
**Presentation:** 3
**Contribution:** 3
**Rating:** 7
**Confidence:** 5

**Summary:**

This paper introduces a spike-based neuromorphic model for sound source localization. It utilizes the RF-PLC methods for auditory-like spectral analysis and encoding. Additionally, it is supported by the MAA module, which simulates attention mechanisms in specific biological frequency bands. These technologies are integrated to enhance the efficiency and accuracy of SSL Tasks.

**Strengths:**

1. The paper describes the RF-PLC method and the MAA module, both of which together replicate the mechanisms of the mammalian auditory pathway, enhancing performance in SSL Tasks.
2. The method is technically sound, with low computational energy consumption, facilitating deployment on neuromorphic computing platforms. In addition,the mathematical proofs are rigorous.
3. The experimental results are comprehensive and reliable, effectively validating the method's effectiveness and robustness.

**Weaknesses:**

1. Regarding the design of the loss function: In practical settings, there should be no significant difference between the model outputs at 355° and 5°. It is worth designing the loss function to avoid this issue effectively. I hope the authors can provide a detailed discussion of the loss function.
2. Concerning robustness validation of localization: To verify the model's accuracy under noisy and complex conditions, the paper involves adding noise to pure audio before classification. I hope the authors can provide a detailed discussion on how the noise dataset was constructed and why it was structured in this way.
3. Validity of experimental methods: The paper appears to be tested only on the SLoClas dataset, which may not sufficiently demonstrate the effectiveness of the proposed method. It is hoped that the authors will test the method on more datasets. If no other public datasets are available, did the authors use any other methods to validate the effectiveness of the proposed approach?

**Questions:**

Please refer to the weaknesses section.

**Limitations:**

Please refer to the weaknesses section.

---

> ### Author Rebuttal · Authors · 2024-08-06
>
> Thank you very much for your valuable feedback and for taking the time to read our paper. We hope the following responses will address your questions.
>
> # W1: How to Design Loss Function
>
> **A**: To achieve a $1\degree$ resolution in SSL tasks, the output layer comprises 360 neurons, each representing a distinct azimuth angle[1]. Subsequently, we use the cross-entropy loss function to guide model training. Thanks to the design of the output layer, there is less need to focus on the discrepancies between the network's predictions and the actual outcomes.
>
> However, during model performance evaluation, the similarity between results at $355\degree$ and $5\degree$ can pose issues in accurately assessing model performance. We utilize the evaluation metrics [2] to valiate the performance of model. Specifically, toaddress the aforementioned issue, **MAE** is further defined as:
>
> $ MAE = \frac{1}{N}\sum_{i=1}^{N} 180\degree -||\hat{\theta}-\theta|-180\degree|$
>
> Reference:
>
> [1] Multi-tone phase coding of interaural time difference for sound source localization with spiking neural networks. In: TASLP (2021).
>
> [2] A hybrid neural coding approach for pattern recognition with spiking neural networks. In TPAMI (2024).
>
> # W2: Robustness Validation of Localization
>
> **A**: Considering the potential presence of noise in SSL tasks, it is crucial to assess model performance in noisy environments. As described in Appendix C, we introduce the same noise information, $noise(n)$, into each microphone channel $mic_i(n)$ to simulate noise coming from various directions. The dataset can be represented as:
>
> $ComplexVideo_i (n) = mic_i (n) + λnoise(n), i=1,2,3,4$
>
> Additionally, we can utilize the Signal-to-Noise Ratio (SNR) to represent the complexity of the dataset after adding noise.
>
> $SNR(dB) = 10 log_{10}(\frac{P_{signal}}{P_{noise}}) $
>
> # W3: Validity of Experimental Methods
>
> **A**: The SLoClas dataset stands as one of the most **challenging** benchmarks for SSL tasks. In contrast, other datasets, such as those employed in studies [1] and [2], suffer from limited localization precision, making them less representative of real-world scenarios. Moreover, we have also tested our model on datasets [1] and [2], where it achieved nearly 100% accuracy. Therefore, we are particularly focused on the performance of the SLoClas dataset, as it is the only one collected in a real-world environment.
>
> |Dataset|Previous SOTA Acc (%)|Ours Acc (%)|Azimuth Range|Resolution|
> |:-:|:-:|:-:|:-:|:-:|
> |HRTF [1]|97.38%|**99.84%**|$-60\degree\sim60\degree$|10$\degree$|
> |Single Words [2]|96.30%|**99.63%**|$-45\degree\sim45\degree$|10$\degree$|
> |**SLoClas [3]**|95.61%|**96.90%**|$-180\degree\sim180\degree$|5$\degree$|
>
> Reference:
>
> [1] Spiking neural network model of sound localization using the interaural intensity difference. In: TNNLS (2012).
>
> [2] A biologically inspired spiking neural network model of the auditory midbrain for sound source localisation. In: Neurocomputing (2010).
>
> [3] A hybrid neural coding approach for pattern recognition with spiking neural networks. In: TPAMI (2024).

---

> > ### Comment · Reviewer_zZvi · 2024-08-12
> >
> > The author answered my confusion and therefore raised the score.

---

> > > ### Author Response · Authors · 2024-08-12
> > > **Thank you for your positive feedback and insightful suggestions.**
> > >
> > > Thank you for your response and recognition. We're pleased that your concerns have been addressed. Your feedback is invaluable in helping us improve the quality of our work.

---

### Official Review · Reviewer_ajui · 2024-07-12

**Soundness:** 2
**Presentation:** 2
**Contribution:** 2
**Rating:** 5
**Confidence:** 2

**Summary:**

This paper introduces a spike-based neuromorphic model designed for sound source localization (SSL), capitalizing on the inherent properties of Resonate-and-Fire (RF) neurons. By encoding sound via phase-locking to leverage the resonance characteristics of these neurons, the model efficiently represents interaural time differences (ITD), facilitating precise sound localization. The inclusion of a multi-auditory attention (MAA) module, inspired by biological auditory systems, enhances the model's performance in noisy conditions by focusing on relevant sound frequencies and temporal sequences. Extensive experimental results demonstrate the model's state-of-the-art accuracy and robustness against noise, presenting an advance in neuromorphic SSL applications.

**Strengths:**

__Comprehensive and Detailed Presentation__: The scheme is presented in full, with detailed descriptions of all modules' design thoughts and operational logics. Ablation experiments validate the effectiveness of each individual improvement.

__Clarity and Empirical Validation__: The paper is well-written and clear, with the experimental section effectively demonstrating key metrics like Parameters, Power, and DoA. These metrics align well with the paper's goals of optimizing energy consumption and performance.

**Weaknesses:**

__Limited Dataset__: The experiments are conducted on a single dataset, which does not validate the universality of the proposed solution.

__Incremental Costs in Module Stacking__: The design seems incremental by stacking modules that introduce additional costs to improve performance. Specifically, the introduction of additional Parameters and Power in the ablation studies (Table 2) appears to contradict the paper's motivation to reduce computing complexity and energy consumption. The trade-off between these aspects may need clarification.

__Novelty__: The application of SNNs for SSL seems to have been first introduced in reference [38]. Thus, the contributions of this paper are optimizations based on [38], and the novelty needs further clarification.

**Questions:**

__Q1: Training Time Cost__. I am curious about the time cost during the training phase. SNNs are known for their training overhead, even though they theoretically have lower power consumption during inference. It might be beneficial for the paper to discuss the additional overhead introduced by the three modules during training, perhaps including this as a metric in the ablation experiments.

__Q2: Suitability of Neuromorphic Systems for SSL Tasks__. Why are neuromorphic systems more suitable for SSL tasks? From Table 1, it appears that the proposed SNN-based solution significantly outperforms ANN-based implementations, which is contrary to the common understanding that ANNs generally lead in performance compared to SNNs. I would appreciate insights from the authors on this discrepancy.

**Limitations:**

See Weaknesses and Questions.

---

> ### Author Rebuttal · Authors · 2024-08-06
>
> Thank you very much for recognizing the strengths and quality contributions of our paper. In response to the questions you raised, we are providing further details and insights to clarify the points mentioned in your review.
> # Q1: Generalization
> **A**: As per your suggestion, we evaluated our model using the HRTF, Single Words, and SLoClas datasets, assembled from both simulations and real-world data collections. As detailed in the subsequent table, our model demonstrates SOTA performance across all the three datasets, thereby validating its generalizability.
>
> |Dataset|Previous SOTA Acc (%)|Ours Acc (%)|Azimuth Range|Resolution|
> |:-:|:-:|:-:|:-:|:-:|
> |HRTF [1]|97.38%|**99.84%**|$-60\degree\sim60\degree$|10$\degree$|
> |Single Words [2]|96.30%|**99.63%**|$-45\degree\sim45\degree$|10$\degree$|
> |**SLoClas [3]**|95.61%|**96.90%**|$-180\degree\sim180\degree$|5$\degree$|
>
> Reference:
>
> [1] Spiking neural network model of sound localization using the interaural intensity difference. In: TNNLS (2012).
>
> [2] A biologically inspired spiking neural network model of the auditory midbrain for sound source localisation. In: Neurocomputing (2010).
>
> [3] A hybrid neural coding approach for pattern recognition with spiking neural networks. In: TPAMI (2024).
>
> # Q2: Incremental Costs in Module Stacking
>
> **A**:  We agree with the importance of balancing performance and computational costs. As per your suggestion, we conducted a detailed comparison across various SSL models. As shown in the following table, our model achieves competitive performance while reducing the model size by 1/10. The results demonstrate that our model achieves the **best trade-offs** between performance and computational overhead.
>
> |Model|Param (M)|MAE ($\degree$)|Acc (%)|
> |:-:|:-:|:-:|:-:|
> |Hybrid Coding [1]|1.61M|0.60$\degree$|95.61%|
> |MTPC-RSNN [2]|1.67M|1.48$\degree$|94.30%|
> |Spike-Driven V2 [3]|15.1M|0.25$\degree$|97.10%|
> |**Ours**|1.64M|0.33$\degree$|96.40%|
>
> Reference:
>
> [1] A hybrid neural coding approach for pattern recognition with spiking neural networks. In: TPAMI (2024).
>
> [2] Multi-tone phase coding of interaural time difference for sound source localization with spiking neural networks. In: TASLP (2021).
>
> [3] Spike-driven Transformer V2: Meta Spiking Neural Network Architecture Inspiring the Design of Next-generation Neuromorphic Chips. In: ICLR (2024).
>
> # Q3: Novelty of This Work
> **A**: Compared to the work referenced in [38], our study demonstrates innovation primarily in two aspects:
> - **Energy-efficient ITD Encoding:** Reference [38] uses FT operations during encoding, which needs massive computing resources. Our method innovatively employs RF spike neurons and phase-locking mechanisms to directly encode raw speech signals into sparse ITD spike trains. This approach is not only biologically plausible but also energy-efficient.
> - **Enhanced Performance in SSL Backend Networks:** The backend network in [38] is limited to convolutional architectures, which do not adequately consider the dependencies and preferences between the frequency and time domains of sound signals, resulting in suboptimal localization performance. In contrast, our MAA module innovatively incorporates the frequency band preferences and temporal interactions found in biological auditory systems. Our ablation studies robustly validate this enhancement.
> # Q4: Training Cost
> **A**: Based on your suggestion, we tested the training time overhead on ANN and SNN backends with identical architectures. Detailed experimental details are as follows:
> - **ANN**: Training was conducted on a single NVIDIA RTX 3090 (24G) with a batch size of 16. Each epoch took approximately **30s**, and the model required around 300 epochs to complete training.
> - **SNN**: Training was conducted on a single NVIDIA RTX 3090 (24G) with a batch size of 16 and a time step of 4. Each epoch took approximately **60s**, and the model required around 200 epochs to complete training.
>
> As you produced, SNNs indeed require longer training times compared to ANNs. Efficiently training SNNs continues to be an unresolved issue within the field. Numerous studies have focused on mitigating the training challenges associated with SNNs through the development of efficient algorithms [1] and the implementation of dedicated hardware acceleration platforms [2,3].
>
> Reference:
>
> [1] Temporal Efficient Training of Spiking Neural Network via Gradient Re-weighting. In: ICLR (2022).
>
> [2] Towards artificial general intelligence with hybrid Tianjic chip architecture. In: Nature (2019).
>
> [3] 22.6 anp-i: A 28nm 1.5 pj/sop asynchronous spiking neural network processor enabling sub-o. 1 μj/sample on-chip learning for edge-ai applications. In: ISSCC (2023).
>
> # Q5: Why Our SSL Model Performance Better
> **A**: This is a core highlight of our work. We attribute the superior performance of our model to the following three key parts:
> - **RF-PLC method**: RF neurons are strategically deployed to precisely isolate ITD cues, effectively eliminating auditory information that does not contribute to localization. This provides a reliable data foundation for high-performance localization.
> - **MAA module**: The MAA module draws on biological auditory principles of frequency band preference and short-term memory. The font enhances networks' focus on critical ITD cues within key frequency bands. The latter strengthens the model's memory for informed decisions across timeframes.
> - **Advantage of SNNs in SSL task**:  Information in SNNs is encoded within temporal spike trains, allowing these networks to directly and effectively capture temporal features. Given the high time resolution required for SSL tasks, SNNs are particularly well-suited for processing temporal audio signals.
>
> Overall, the proposed SSL model well-orchestrates the proposed RF-PLC method, MAA module and the inherent advantage of SNNs, and this is the main reason for its high performance.

---

### Official Review · Reviewer_2syp · 2024-07-13

**Soundness:** 2
**Presentation:** 2
**Contribution:** 1
**Rating:** 4
**Confidence:** 4

**Summary:**

This work proposes a SNN-based model for SSL. To achieve efficient processing of raw
 speech signals, they introduce a phase-locking coding (RF-PLC) method using Resonate-and-Fire (RF)
 neurons and detection neurons.

**Strengths:**

This work proposes a SNN-based model for SSL. To achieve efficient processing of raw
 speech signals, they introduce a phase-locking coding (RF-PLC) method using Resonate-and-Fire (RF)
 neurons and detection neurons.

**Weaknesses:**

1. The improvements of performance compared with current SOTA methods are quite limited. And the paper did not compred with Spike-driven V2 and other SOTA methods. Hence the contribution on model performance improvements  is unclear.

2. The novelty of proposed attention mechanisms  is limited compared to existing  attention methods for SNN.

3. The experiments results should be conducted multiple trials. The source code could be provided to show the reproducability.

**Questions:**

1. The improvements of performance compared with current SOTA methods are quite limited. And the paper did not compred with Spike-driven V2 and other SOTA methods. Hence the contribution on model performance improvements  is unclear.

2. The novelty of proposed attention mechanisms  is limited compared to existing  attention methods for SNN.

3. The experiments results should be conducted multiple trials. All source code could be provided to show the reproducability.

**Limitations:**

See mentioned above.

---

> ### Author Rebuttal · Authors · 2024-08-06
>
> Thank you very much for your review comments. In response to the issues you have raised, we offer the following replies:
>
> # Q1: Limited Improvement
> - **Limited Improvement**: In SSL tasks, Acc and MAE are the most important metrics. As shown in the following table, extensive comparative experiments have demonstrated that our model not only achieves SOTA accuracy among similarly sized models but also reduces MAE by approximately **50%**.
>
> |Methods|Type|Model Size|MAE ($\degree$)|Acc (%)|
> |:-:|:-:|:-:|:-:|:-:|
> |GCC-PHAT-CNN [1]|ANN|4.17M|4.39$\degree$|86.94%|
> |SELDnet [2]|ANN|1.68M|1.78$\degree$|88.24%|
> |EINV2 [3]|ANN|1.63M|0.98$\degree$ |94.64%|
> |SRP-DNN [4]|ANN|1.64M|0.96$\degree$|94.12%|
> |MTPC-CSNN [5]|SNN|1.61M|1.23$\degree$|93.95%|
> |MTPC-RSNN [5]|SNN|1.67M|1.48$\degree$|94.30%|
> |Hybrid Coding [6]|SNN|1.61M|0.60$\degree$|95.61%|
> |**Ours**|SNN|1.64M|**$0.33\degree$**|**96.40%**|
>
> Reference can be found in Table.1 of the submitted manuscript.
>
> - **Compared with Spike-Driven V2 Model**:   As per your suggestion,  we replaced the back-end model with Spike-driven V1 and V2 models [1,2] for the ablation study.  As shown in the following table, our model achieves competitive performance while reducing the model size by **1/10**. Therefore, it achieves the **best trade-offs** between performance and computational costs.
>
> | Model | Param (M) | MAE ($\degree$) |
> |:-----:|:---------:|:----------:|
> | Spike-Driven [1] | 16.81M | $0.27\degree \pm 0.05\degree$ |
> | Spike-Driven V2 [2] | 15.1M | $0.25\degree \pm 0.04\degree$ |
> |  **Ours**  | **1.64M** | $0.33\degree \pm 0.02\degree$ |
>
> By the way,  spike-driven methods are generic and high-performing, but they rely on extensive parameters. In contrast, our model is specifically designed for SSL tasks, fully considering the characteristics of ITD pulse sequences, thus it is both lightweight and efficient.
>
> Reference:
>
> [1] Spike-driven Transformer. In: NeurIPS (2023).
>
> [2] Spike-driven Transformer V2: Meta Spiking Neural Network Architecture Inspiring the Design of Next-generation Neuromorphic Chips. In: ICLR (2024).
>
> # Q2: Novelty of Attention Mechanisms
>
> **A**: The novelty of our MAA module lies in three key aspects:
> - **MAC-free Computational Paradigm:** Compared with those attention methods [1,2,3] in SNNs, our methods achieve MAC-free attention computation paradigm. Therefore, it possesses significant energy efficiency advantages.
>
> - **Task-Specific Design:**  Our methods are designed to work well with the specific patterns of ITD spike trains from the RF-PLC method. This helps the front-end and back-end of the SSL model work better together, making the system both efficient and effective.
>
> - **Biological Plausibility:** The MAA incorporates frequency band preference and short-term memory, both grounded in strong biological evidence. Neuroscientific evidence indicates that frequency tuning varies across regions, highlighting the critical role of frequency selectivity in auditory processing [4]. Additionally, biological studies demonstrate that auditory short-term memory significantly enhances selective attention to auditory signals in noisy environments [5].
>
> Reference:
>
> [1] Temporal-wise attention spiking neural networks for event streams classification. In: ICCV (2021).
>
> [2] TCJA-SNN: Temporal-channel joint attention for spiking neural networks. In: TNNLS (2024).
>
> [3] Attention spiking neural networks. In: TPAMI (2023).
>
> [4] Temporal coherence and attention in auditory scene analysis. In: TRENDS NEUROSCI (2011).
>
> [5] Attention improves memory by suppressing spiking-neuron activity in the human anterior temporal lobe. In: Nature Neuroscience (2018).
> # Q3: Experiment Detail
>
> **A**:  As per your suggestions, all experiments in this paper were conducted **at least 5 times** to ensure reliability. For example, the follow table shows the robustness performance of our methods in five independent experiments.  In addition, **we have uploaded all the code to the supplementary materials to help others replicate our results more easily**.
>
> |SNR (dB)|MAE ($\degree$)|Acc (%)|
> |:-:|:-:|:-:|
> |50 dB (Low)|$0.33\degree \pm 0.02\degree$|$96.40$\% $\pm$ $0.30$\%|
> |20 dB (Middle)|$0.43\degree \pm 0.03\degree$|$95.60$\% $\pm$ $0.23$\%|
> |0 dB (High)|$0.54\degree \pm 0.04\degree$|$94.90$\% $\pm$ $0.15$\%|

---

### Official Review · Reviewer_PWNz · 2024-07-18

**Soundness:** 2
**Presentation:** 1
**Contribution:** 2
**Rating:** 3
**Confidence:** 5

**Summary:**

The paper presents a novel neuromorphic model for sound source localization (SSL) inspired by biological auditory systems. The model integrates spike-based neural encoding and computation, employing Resonate-and-Fire (RF) neurons with a phase-locking coding (RF-PLC) method. The RF-PLC method leverages the resonance properties of RF neurons to efficiently convert audio signals to time-frequency representations and encode interaural time difference (ITD) cues into discriminative spike patterns. Additionally, the model incorporates a spike-driven multi-auditory attention (MAA) module inspired by biological adaptations, which enhances SSL capability in noisy environments. The authors demonstrate that their model achieves state-of-the-art accuracy and robustness in real-world conditions.

**Strengths:**

Biological Inspiration: The model is well-motivated by biological mechanisms, which is a strong point given the effectiveness of biological systems in sound localization. Energy Efficiency: The use of RF neurons and phase-locking coding is innovative and contributes to the energy efficiency of the model. Robustness: The introduction of the MAA module significantly improves robustness and accuracy in noisy environments. Performance: The experimental results are promising, showing that the model achieves state-of-the-art accuracy and maintains high performance even at low signal-to-noise ratios.

**Weaknesses:**

Comparative Analysis: While the paper claims state-of-the-art performance, the comparative analysis with existing models is not comprehensive. The authors should provide more detailed comparisons and discuss why their model outperforms others. Complexity: The model's complexity and computational requirements are not adequately addressed. It's important to understand the trade-offs between performance gains and computational costs. Generalization: The experiments are promising but limited in scope. Additional tests across various datasets and real-world scenarios are needed to validate the generalizability of the model. Biological Plausibility: The biological plausibility of some components, such as the MAA module, is not thoroughly discussed. More insights into how closely these components mimic biological processes would strengthen the paper.

**Questions:**

The introduction provides a clear motivation and context for the research. However, it could benefit from a more detailed discussion of existing SSL models and their limitations. The methodology is well-described, but some parts are overly technical and could be simplified for clarity. Diagrams illustrating the RF-PLC method and MAA module would be helpful.The results section presents compelling evidence of the model's performance. Including statistical significance tests and error bars in the figures would add rigor to the findings. The discussion touches on the implications of the results but lacks depth in exploring future work and potential applications. More emphasis on the practical applications of the model would be beneficial.

**Limitations:**

See questions.

---

> ### Author Rebuttal · Authors · 2024-08-06
>
> We greatly appreciate your recognition of the innovative aspects and motivation of our work. In response to the weaknesses and questions, we will provide further detailed explanations:
>
> # W1: Compare Analysis
> **A**: Following your suggestion , we rigorously evaluated our work against previous SOTA ANN-based and SNN-based models in terms of both Acc and MAE. As demonstrated in the table, our model all achieved SOTA performance.
> |Methods|Type|MAE ($\degree$)|Acc (%)|
> |:-:|:-:|:-:|:-:|
> |GCC-PHAT-CNN [1]|ANN|4.39$\degree$|86.94%|
> |SELDnet [2]|ANN|1.78$\degree$|88.24%|
> |EINV2 [3]|ANN|0.98$\degree$ |94.64%|
> |SRP-DNN [4]|ANN|0.96$\degree$|94.12%|
> |MTPC-CSNN [5]|SNN|1.23$\degree$|93.95%|
> |MTPC-RSNN [5]|SNN|1.48$\degree$|94.30%|
> |Hybrid Coding [6]|SNN|0.60$\degree$|95.61%|
> |**Ours**|SNN|**$0.33\degree$**|**96.40%**|
>
> We attribute the superior performance of our model to the following two key parts:
> - **RF-PLC method**: RF neurons are strategically deployed to precisely isolate ITD cues, effectively eliminating auditory information that does not contribute to localization. This provides a reliable data foundation for high-performance localization.
> - **MAA module**: The MAA module draws on biological auditory principles of frequency band preference and short-term memory. The font enhances networks' focus on critical ITD cues within key frequency bands. The latter strengthens the model's memory for informed decisions across timeframes.
>
> Overall, our SSL model effectively integrates the proposed RF-PLC method and MAA module, resulting in demonstrably superior performance.
>
> # W2: Trade-offs between Performance and Computational Costs
> **A**: As per your suggestion, we conducted a detailed comparison across various SSL models. As shown in the following table, our model achieves competitive performance while reducing the model size by **1/10**. The results demonstrate that our model achieves the **best trade-offs** between performance and computational overhead.
> |Model|Param (M)|MAE ($\degree$)|Acc (%)|
> |:-:|:-:|:-:|:-:|
> |Hybrid Coding [1]|1.61M|0.60$\degree$|95.61%|
> |MTPC-RSNN [2]|1.67M|1.48$\degree$|94.30%|
> |Spike-Driven V2 [3]|15.1M|0.25$\degree$|97.10%|
> |**Ours**|1.64M|0.33$\degree$|96.40%|
>
> Reference:
>
> [1] A hybrid neural coding approach for pattern recognition with spiking neural networks. In: TPAMI (2024).
>
> [2] Multi-tone phase coding of interaural time difference for sound source localization with spiking neural networks. In: TASLP (2021).
>
> [3] Spike-driven Transformer V2: Meta Spiking Neural Network Architecture Inspiring the Design of Next-generation Neuromorphic Chips. In: ICLR (2024).
>
> # W3: Generalization
> **A**: As per your suggestion, we evaluated our model using the HRTF, Single Words, and SLoClas datasets, assembled from both simulations and real-world data collections. As detailed in the subsequent table, our model demonstrates SOTA performance across these datasets, thereby validating its generalizability.
>
> |Dataset|Previous SOTA Acc (%)|Ours Acc (%)|Azimuth Range|Resolution|
> |:-:|:-:|:-:|:-:|:-:|
> |HRTF [1]|97.38%|**99.84%**|$-60\degree\sim60\degree$|10$\degree$|
> |Single Words [2]|96.30%|**99.63%**|$-45\degree\sim45\degree$|10$\degree$|
> |**SLoClas [3]**|95.61%|**96.90%**|$-180\degree\sim180\degree$|5$\degree$|
>
> Reference:
>
> [1] Spiking neural network model of sound localization using the interaural intensity difference. In: TNNLS (2012).
>
> [2] A biologically inspired spiking neural network model of the auditory midbrain for sound source localisation. In: Neurocomputing (2010).
>
> [3] A hybrid neural coding approach for pattern recognition with spiking neural networks. In: TPAMI (2024).
>
> # W4: Biological Plausibility
> **A**: The biological plausibility of our model is demonstrated in two key aspects:
> - **RF-PLC method**: This method utilizes RF spiking neurons and phase-locking mechanisms, both of which are biologically plausible. Neuroscientific evidence demonstrates that RF neurons, a specific type of biological neuron, selectively respond to particular input frequencies and efficiently process oscillatory signals [1]. Furthermore, phase-locking mechanisms are prevalent in the biological auditory system and facilitate the capture and encoding of temporally correlated information by neurons [2].
> - **MAA module**: The MAA module emphasizes frequency band preference and short-term memory. Neuroscientific research indicates the critical role of frequency selectivity in auditory processing [3]. Additionally, biological studies demonstrate that short-term memory enhances selective auditory attention in noisy environments [4].
>
> Reference:
>
> [1] Dendritic channelopathies contribute to neocortical and sensory hyperexcitability in Fmr1−/y mice. In: NAT NEUROSCI (2014).
>
> [2] An oscillator model better predicts cortical entrainment to music. In PANS (2019).
>
> [3] Temporal coherence and attention in auditory scene analysis. In: TRENDS NEUROSCI (2011).
>
> [4] Attention improves memory by suppressing spiking-neuron activity in the human anterior temporal lobe. In: NAT NEUROSCI (2018).
> # Q1: Suggestions for Improving Writing
> **A**: Thank you for your constructive feedback on our manuscript. We will carefully consider your recommendations to simplify technical descriptions and enhance discussions of limitations relative to existing SSL models.
> # Q2: Experimental Reproducibility and Future Discussion
> **A**: As shown in the following table, all experiments were conducted **at least 5 times** to ensure reliability.Furthermore, **the code has been uploaded to the supplementary materials to enhance the reproducibility**. In future research, we aim to deploy and test our model on Loihi 2  or Tianjic, ultimately applying it to intelligent robots.
> |SNR (dB)|MAE ($\degree$)|Acc (%)|
> |:-:|:-:|:-:|
> |50 dB (Low)|$0.33\degree \pm 0.02\degree$|$96.40$\% $\pm$ $0.30$\%|
> |20 dB (Middle)|$0.43\degree \pm 0.03\degree$|$95.60$\% $\pm$ $0.23$\%|
> |0 dB (High)|$0.54\degree \pm 0.04\degree$|$94.90$\% $\pm$ $0.15$\%|

---

> ### Comment · Reviewer_PWNz · 2024-08-13
> **Comments**
>
> Thanks for rebuttal, but the results seem not convinced just as reviewer 2syp mentioned, so I will lower my score.

---

> > ### Author Response · Authors · 2024-08-13
> >
> > Dear Reviewer,
> >
> > We sincerely appreciate your response. In terms of the convincing results problem pointed by the Reviewer 2syp in Q3,  we have included all relevant codes and data in the supplementary materials alongside our original manuscript. In our work, all the experiments in our work are conducted at least five times. We apologize for any misunderstanding this may have caused and kindly request that you review the supplementary materials to verify the performance of our methods.
> >
> > Moreover, extensive comparative experiments have demonstrated that our model not only achieves SOTA accuracy among similarly sized models but also reduces MAE by approximately 50%. Especially, compared to current SOTA models, such as Spike-driven V2, our model is significantly smaller, with its size being only 1/10 of theirs.
> >
> > We hope our responses have addressed your concerns. We look forward to your further comments and evaluation.

---

> ### Author Response · Authors · 2024-08-13
> **Seeking Comments on Updated Review Scores**
>
> Thank you for your thoughtful review and valuable suggestions regarding our manuscript. We have carefully incorporated your recommendations and conducted extensive experiments to address your concerns.
>
> We observed that the score for our manuscript decreased from 4 to 3. We are puzzled by this change and would like to understand the reasons. Could this reduction be a misunderstanding, or are there new concerns arising from our rebuttal? We would greatly appreciate any additional feedback that could help us improve our manuscript further.
>
> Thank you very much for considering our request.

---

> ### Author Response · Authors · 2024-08-14
>
> Dear Reviewer,
>
> We would like to confirm whether our recent response has adequately addressed your concerns. Regarding the issue of convincing results, all the relevant codes and data are provided in the supplementary materials. May we kindly request that you review these materials at your earliest convenience? **We appreciate your reconsideration and hope for a fair evaluation**.
>
> Thank you for your time and attention.

---

### Author Response · Authors · 2024-08-12
**Request for Feedback on Response Clarifications**

Dear Reviewers,

Could you please provide feedback on our responses to your questions? We are eager to know if our answers have addressed your concerns. Additionally, are there any further comments or insights you might have regarding our work?

Your input is crucial for us to enhance the quality of our submission, and we greatly appreciate your guidance and time.

---

### Decision · Program_Chairs · 2024-09-25

**Decision:**

Accept (poster)

**Comment:**

This paper presents a novel Sounds Source Localization (SSL) model that is based on Spiking Neural Networks (SNNs). Specifically, the paper use RF-PLC neurons to encode interaural time differences that are critical to SSL. These neurons avoid the significant computational overhead of calculating Fourier transforms as used in state-of-the-art (SOTA) SSL baselines. Additionally, the authors propose a SNN based multi-auditory attention module which implements biologically inspired properties of frequency selectivity and short-term auditory memory, which is known to improve SSL accuracy and robustness in noisy environments. The paper presents experiments demonstrating the superior accuracy and robustness of the proposed SNN-SSL on SloClas dataset. Analytical energy estimates show that the proposed SSL model has comparable energy to the SOTA SNN baseline.

The reviewers primarily suggested adding more datasets and more SNN/ANN SSL baselines to compare the proposed model against. The authors have done a great job by adding SSL accuracy data for more datasets and other baselines and all experiments continue to show strong accuracy results for the proposed SSL. Spike-Driven V2 is one baseline that achieves a better accuracy than this paper, but as the authors highlight, it also has ~10x more parameters compared to the proposed model. Consequently, the proposed SSL offers a better tradeoff when it comes to accuracy vs model size.

The authors showed great flexibility by adding ablation experiments that further strengthened the justification for their model design choices. To conclude, the paper presents a novel SSL model which is biologically motivated and shows high accuracy and robustness for the given model size. I recommend that this paper should be accepted for NeurIPS.

Minor note, the authors should proof-read their submission for small grammatical errors and try to organize the paper to improve its readability. For e.g., Table 2 shows useful ablation study data, but it could use a more detailed description describing all the experiment configs. Also, the authors show comparable energy consumption to the existing SNN-SSL baseline in Table 2, adding a comment on that in the results section of the paper will be helpful.

Edit: Updated my recommendation to poster (prev spotlight) post discussion with SAC.